# ACTIVATION DECAY BY LOSS SMOOTHING TO ENHANCE GENERALIZATION

## ABSTRACT

Generalization in deep learning is often associated with the sharpness of the minima encountered during training. We introduce a novel, deterministic, and computationally efficient method called *activation decay*, designed to flatten sharp minima and improve generalization across a wide range of tasks. Derived from Gaussian smoothing, activation decay operates by regularizing the activations of critical network layers, effectively reducing sharpness and improving robustness. Unlike stochastic techniques such as dropout or the more computationally expensive Sharpness-Aware Minimization (SAM), our approach requires no additional computational overhead, making it particularly suited for large-scale models. We further demonstrate that activation decay can be seamlessly combined with other regularization techniques, offering enhanced regularization without increasing training complexity. Extensive experiments on CIFAR-10, ImageNet, and natural language processing (NLP) tasks validate our approach, showing consistent improvements in generalization and robustness to label noise.

## 1 INTRODUCTION

Generalization in deep learning models remains a fundamental challenge, particularly as models grow in complexity and are tasked with learning from increasingly large datasets. The nature of the minima in the loss landscape has often been cited as a key factor influencing generalization. Models that converge to sharp minima tend to be highly sensitive to small perturbations, resulting in poor performance on unseen data. In contrast, models that converge to flatter minima have often been associated with improved robustness and generalization (Hochreiter & Schmidhuber, 1997). While sharpness-based measures have been shown to correlate strongly with generalization under certain settings (Jiang et al., 2020), it is important to note that flatness is not a universal guarantee of better generalization, as it is sensitive to parameterization and re-scaling effects in deep, overparameterized networks (Zhang et al., 2017). Despite these nuances, regularization techniques aimed at minimizing sharpness or curvature remain effective heuristics in guiding models toward solutions that generalize well in practice.

A widely used regularization method is weight decay, or $\ell_2$ regularization on weights, which penalizes the magnitude of the model's weights by adding a term proportional to the squared norm of the parameters to the loss function. Weight decay penalizes large parameter values, smoothing the loss landscape and reducing the complexity of the learned model. By controlling the magnitude of the weights, weight decay helps the model find flatter minima, improving generalization (Krogh & Hertz, 1992).

Bishop (1995) first established the connection between noise injection and deterministic regularization, demonstrating how noise injection can be cast as a form of regularization. In more recent work, Wei et al. (2020); Orvieto et al. (2022) further explored this connection by analyzing how specific forms of noise injection, such as anticorrelated noise and dropout (Srivastava et al., 2014), can encourage flatter minima and improve generalization. While noise injection introduces stochasticity during training, making models more robust to perturbations and guiding them toward flatter minima, those methods introduce randomness into the optimization process, which can lead to performance variability across different runs.

Another important method in the field of regularization is *Sharpness-Aware Minimization* (SAM) (Foret et al., 2021), which directly targets sharpness by introducing a min-max optimization frame-

work. SAM aims to minimize the worst-case sharpness in a neighborhood of the model's parameters, encouraging convergence to flatter minima. While effective, SAM's computational cost and memory overhead are significantly higher than simpler methods like weight decay and noise injection. Zhang et al. (2018) show that batch normalization (Ioffe & Szegedy, 2015) and residual connections (He et al., 2016) enhance backpropagation by improving the local Hessian's spectrum leading to better gradient flow.

Beyond simply smoothing the loss landscape, methods like SAM provide robustness to label noise by guiding models to converge to flatter minima, where the sensitivity to noisy labels is reduced (Baek et al., 2024). Similarly, smooth networks, such as Lipschitz networks, demonstrate robustness to adversarial attacks (Tsuzuku et al., 2018; Cohen et al., 2019), as controlling the Lipschitz constant ensures that small perturbations in the input lead to bounded changes in the output. Importantly, the Lipschitz constant of a network also plays a critical role in generalization bounds (Bartlett et al., 2017), as it governs how sensitive the model is to variations in the data, further linking generalization performance with robustness to both label noise and adversarial perturbations.

The main contributions of this paper are as follows:

- We provide a theoretical framework that establishes a clear relationship between noise variance and the spectral norm of the Hessian. Specifically, we present two results: Corollary 1 quantifies the reduction in Hessian curvature due to Gaussian smoothing near a local minimum, offering guarantees on how smoothing influences the loss landscape and improves generalization. Theorem 1 demonstrates how the curvature of the final layer and loss, along with the Lipschitz constants of the intermediate layers, regularizes the Hessian by decomposing the individual contributions of each layer to the overall Hessian.

- We propose a simple and deterministic method based on Gaussian smoothing applied to the final layer weights in Theorem 2. This method significantly reduces the computational overhead while retaining the smoothing benefits with regard to flat minima. We call that method *activation decay* (AD) as it can be cast as a $\ell_2$-norm regularization on activations.

- We validate our approach through extensive experiments on CIFAR-10, ImageNet, and natural language processing (NLP) benchmarks in a multi-task setting. Our method consistently improves generalization, demonstrates robustness to label noise (on CIFAR-10), and outperforms SAM and dropout on NLP tasks. Additionally, it can be seamlessly combined with SAM without additional computational overhead.

The paper is organized as follows: Section 2 reviews the related work on regularization techniques in neural networks to promote flat minima, focusing on approaches such as noise-based regularization, $\ell_2$ regularization, and their impact on generalization. Section 3 presents our theoretical framework, detailing the relationship between Gaussian smoothing, Hessian curvature, and $\ell_2$ regularization, including new insights into how layer-wise regularization impacts the overall loss landscape. Section 4 describes our experimental setup, discusses the results, and demonstrates the effectiveness of our approach across multiple tasks.

## 2 BACKGROUND & RELATED WORK

We define a feed-forward neural network as follows: $\boldsymbol{h}^{(L)} = \mathbf{W}^{(L)}\boldsymbol{h}^{(L-1)}$ where $\boldsymbol{h}^{(l-1)} = f^{(l-1)}(\boldsymbol{h}^{(l-2)}) = s^{(l-1)}(\mathbf{W}^{(l-1)}\boldsymbol{h}^{(l-2)})$ for layers $l = 1, \ldots, L-1$ and $\boldsymbol{h}^{(0)} = \boldsymbol{x}$ denotes the input data. Each layer $l$ has its activation $s^{(l)}$, typically ReLU or GELU. For the sake of simplicity, we omit bias in the network. We note $\|\boldsymbol{W}^{(l)}\|_2$ the spectral norm of matrix $\boldsymbol{W}^{(l)}$. For a given label $\boldsymbol{y}$, we denote the loss as $\mathcal{L}(\boldsymbol{h}^{(L)}, \boldsymbol{y}, \boldsymbol{\theta})$. The collection of all parameters to be learned, $\boldsymbol{\theta}$, include all the weights in the network: $\boldsymbol{\theta} = \text{vec}\left(\{\mathbf{W}^{(l)}\}_{l=1,\ldots,L}\right)$. We refer to the loss function as $\mathcal{L}(\boldsymbol{\theta})$ for brevity.

Regularization techniques are essential for enhancing the generalization capability of neural networks by controlling model complexity, reducing overfitting, and stabilizing training.

## 2.1 FLAT MINIMA

Generalization, the ability of a model to perform well on unseen data, is often associated with the flatness of the minima found during training, as highlighted by empirical studies (Keskar et al., 2016; Chaudhari et al., 2019), where sharp minima are linked to poorer generalization. The relationship between loss landscapes, generalization, and stochastic gradient descent (SGD) has been a central topic in machine learning research for years (Hochreiter & Schmidhuber, 1997). For instance, it has been shown that in overparameterized models, local minima of the loss function are often close to the global minima (Choromanska et al., 2015). Additionally, Xing et al. (2018) demonstrate that SGD exhibits an implicit bias, favoring regions of the loss landscape resembling a valley.

A pivotal study by Keskar et al. (2016) shows that large-batch training tends to converge to sharp minima, which correlates with worse generalization compared to the flatter minima achieved with small-batch training. However, Dinh et al. (2017) point out that common sharpness metrics, such as the spectral norm of the Hessian of $\mathcal{L}(\boldsymbol{\theta})$, are sensitive to re-scaling. In our analysis, the networks are not adversarially reparameterized, ensuring a fair comparison. It is important to note that flatter minima do not always guarantee better generalization, as counterexamples exist where sharp minima generalize well and flat minima perform poorly (Andriushchenko & Flammarion, 2022; Zhang et al., 2017). Nevertheless, Jiang et al. (2020) show that sharpness-based metrics often outperform other complexity metrics for evaluating generalization.

Finally, as discussed in Section 2.3, Sharpness-Aware Minimization (SAM), introduced by Foret et al. (2021), has become a prominent and deeply studied method in this area.

## 2.2 NOISE INJECTION

Noise injection methods add stochasticity during training, either to activations or weights. It improves robustness and prevents overfitting. The most widely known is probably the *dropout* (Srivastava et al., 2014), which randomly drops units during training, reducing co-adaptation of neurons and improving generalization. However, its stochastic nature introduces additional variance in performance, requiring careful tuning to prevent underfitting, particularly when applied to deep models (Liu et al., 2023). The increased training variability may also result in unstable training dynamics under certain conditions. Perturbed Gradient Descent (PGD) (Jin et al., 2017) introduces noise into the weight updates to help models escape saddle points and better explore the optimization landscape. Building on PGD, anticorrelated noise (Orvieto et al., 2022) modifies the noise structure and proves it promotes convergence to flatter minima by controlling the curvature of the loss landscape. The resulting optimization program can be summarized with the smoothed loss: $\mathcal{L}^{\sigma}(\boldsymbol{\theta}) = \mathbb{E}_{\boldsymbol{\Delta} \sim \mathcal{N}(0, \sigma^2 I)} [\mathcal{L}(\boldsymbol{\theta} + \boldsymbol{\Delta})]$. The resulting optimization landscape becomes smoother, reducing the impact of sharp regions. However, tuning the noise parameter $\sigma$ is critical, as improper choices can lead to instability with exploding variance or insufficient regularization.

## 2.3 LOSS SMOOTHING AND REGULARIZATION

While noise-based regularization techniques have been widely adopted, they often introduce undesirable variance into the training process, which can hinder performance. The work of Bishop (1995) interprets Gaussian noise on inputs as Tikhonov regularization on parameters also known as weight decay, establishing a connection between noise injection and deterministic loss smoothing. This encourages smaller weights and promotes simpler solutions that generalize better (Krogh & Hertz, 1992). Such regularization flattens the loss landscape, leading to convergence toward wider, flatter minima that correlate with improved generalization (Hochreiter & Schmidhuber, 1997). The regularized loss function can be expressed as: $\mathcal{L}(\boldsymbol{\theta}) + \frac{\sigma^2}{2} \|\boldsymbol{\theta}\|_2^2$.

Sharpness-Aware Minimization (SAM) (Foret et al., 2021), a more recent approach, penalizes sharp minima directly. SAM minimizes a robust smoothed objective: $\min_{\boldsymbol{\theta}} \max_{\boldsymbol{\Delta} \in \mathcal{B}(\mathbf{0}, \rho)} \mathcal{L}(\boldsymbol{\theta} + \boldsymbol{\Delta})$, where $\mathcal{B}(0, \rho)$ represents a ball of radius $\rho$ around zero, and $\boldsymbol{\Delta}$ is the perturbation applied to the parameters. By introducing an adversarial component into the optimization process, SAM effectively reduces sharpness in the loss landscape, which improves generalization across many tasks. However, SAM's computational cost is significantly higher due to multiple forward-backward passes required for the perturbed loss evaluation.

Notable works on layer wise regularization such as margin constraint Elsayed et al. (2018) or activation regularization Yashwanth et al. (2024) are close to our approach. Inspired by the works of Bishop (1995) and Orvieto et al. (2022), we propose a novel deterministic noise-based regularization that operates on activations rather than weights, with the same low computational cost. This approach reduces sharpness in the loss landscape while maintaining computational efficiency.

## 3 ACTIVATION DECAY: CONTROLLING THE HESSIAN NORM TO IMPROVE GENERALIZATION

In this section, we provide a theoretical framework to quantify how smoothing techniques, such as Gaussian noise injection, affect the curvature of the loss landscape and contribute to better generalization. Specifically, we examine how these techniques influence the spectral norm of the Hessian, a key measure of sharpness in the loss function. By doing so, we establish a connection between noise-based regularization and the flattening of minima. This is followed by an analysis of how constraining the spectral norm weight and the Hessian of individual layers can further control the overall curvature, offering a tractable approach to loss smoothing. Finally, we discuss a practical method of smoothing the loss through activation decay, providing theoretical guarantees for its effectiveness in reducing sharpness. All proofs are provided in the Appendix.

### 3.1 LOSS FLATTENING AND LAYER SMOOTHING

In the regime near a minimum, where the loss gradient's norm is small, understanding the impact of smoothing techniques like Gaussian noise on the curvature of the loss is crucial for generalization. The following corollary quantifies the reduction in the spectral norm of the Hessian—an indicator of sharpness—when Gaussian noise is applied to the parameters near a local minimum. This provides a concrete measure of how smoothing leads to flatter minima, which is directly tied to improved generalization in deep learning.

**Corollary 1** (Dimension-free bound on Hessian norm of Gaussian smoothed loss). *(Adapted from Delattre et al. (2024)) Let $\mathcal{L} : \mathbb{R}^d \to \mathbb{R}$ be differentiable. Suppose that the gradient $\nabla_\theta \mathcal{L}$ is $H$-Lipschitz continuous, and that $\|\nabla_\theta \mathcal{L}(\theta)\|_2 \leq \epsilon$. The spectral norm of the Hessian of the Gaussian smoothed loss is bounded by:*

$$\|\nabla_{\boldsymbol{\theta}}^2 \mathbb{E}_{\boldsymbol{\Delta} \sim \mathcal{N}(0, \sigma^2 I)} [\mathcal{L}(\boldsymbol{\theta} + \boldsymbol{\Delta})]\|_2 \leq H \operatorname{erf}\left(\frac{\epsilon}{\sqrt{2}H\sigma}\right) . \tag{1}$$

In this corollary, $\operatorname{erf}$ denotes the error function defined for any positive and real value by $\operatorname{erf}(x) = \frac{2}{\sqrt{\pi}} \int_0^x e^{-t^2} dt$. The proof of this corollary is derived and adapted from Theorem 2 of Delattre et al. (2024) which is used in randomized smoothing context with noise on inputs to derive Lipschitz continuity of smoothed classifier. The work of Nesterov & Spokoiny (2017) was the first to derive bounds on the regularity of the Gaussian smoothed loss. However, their bound depends on the dimension $d$, which makes it less tight in high-dimensional settings.

This bound offers a quantitative measure of the effect of Gaussian noise injection on the curvature of the loss around minima, providing practical insight into how parameter perturbations contribute to smoother loss landscapes. Note that Orvieto et al. (2022) established a link between the trace of the Hessian of the loss and the noise injected whereas the bound here is on the Hessian's spectral norm.

While directly controlling the overall curvature of the loss landscape is crucial, managing the spectral norm of intermediate layers $\|\boldsymbol{W}^{(l)}\|_2$ and the Hessian of specific layers, such as the loss function, offers a more tractable approach. The next result provides a decomposition of the contributions of each layer, showing how constraining the Lipschitz constants and the Hessian at critical points can effectively control the overall curvature.

**Theorem 1** (Bound on Hessian norm of loss). *Let $\mathcal{L} : \mathbb{R}^d \to \mathbb{R}$ be twice differentiable. Assume that the set of parameters $\boldsymbol{\theta}$ is such that the loss function attains zero: $\mathcal{L}(\boldsymbol{\theta}) = 0$.*

*Then, the spectral norm of the Hessian of the loss $\mathcal{L}$ is bounded by:*

$$\|\nabla^2_{\boldsymbol{\theta}}\,\mathcal{L}(\boldsymbol{\theta})\|_2 \leq \left(\sum_{j=1}^{L-1}\left\|\frac{\partial \mathbf{h}^{(j)}}{\partial \boldsymbol{\theta}}\right\|_2 \prod_{l=j+1}^{L-1}\left\|\mathbf{W}^{(l)}\right\|_2\right)^2 \left\|\nabla^2_{\mathbf{h}^{(L-1)}}\mathcal{L}(\boldsymbol{\theta})\right\|_2 . \tag{2}$$

In over-parameterized deep neural networks, it is common for the training loss to reach values close to zero, given the high capacity of the model to fit the training data (Zhang et al., 2017). Therefore, assuming $\mathcal{L}(\boldsymbol{\theta}) = 0$ is reasonable in this context.

This inequality demonstrates that the contributions from the later layers to the bound on the Hessian norm are more significant than those from the earlier layers, as the spectral norm $\left\|\boldsymbol{W}^{(l)}\right\|_2$ are accumulated more times in deeper layers. Consequently, the spectral norm constants associated with deeper layers influence the curvature of the loss more strongly. The term $\left\|\nabla^2_{\boldsymbol{h}^{(L-1)}}\mathcal{L}(\boldsymbol{\theta})\right\|_2$ reflects the influence of the Hessian with respect to the penultimate activations, which, combined with the accumulated Lipschitz constants, governs the overall smoothness of the loss landscape.

Weight decay is an efficient regularization because it helps to bound spectral norm $\left\|\boldsymbol{W}^{(l)}\right\|_2$ by bounding the Frobenius norm of the layer weights $\left\|\boldsymbol{W}^{(l)}\right\|_F$. The work of Delattre et al. (2023) introduces a tight bound on the spectral norm of convolutional and dense layers, and performs regularization to better generalization.

### 3.2 ACTIVATION DECAY FOR DETERMINISTIC LOSS SMOOTHING

In this section, we implement a Gaussian-smoothed loss function aimed at reducing the sharpness of minima encountered during training by targeting the term $\nabla^2_{\boldsymbol{h}^{(L-1)}}\mathcal{L}(\boldsymbol{\theta})$: we focus on injecting Gaussian noise into the weights $\boldsymbol{W}^{(L)}$ of the final layer $\boldsymbol{h}^{(L)}$ while keeping the rest of the network unchanged. Perturbing only the final layer, we reduce computational overhead while retaining the smoothing benefits. The smoothed final loss can be written as:

$$\mathcal{L}^{\sigma}(\boldsymbol{W}^{(L)}\boldsymbol{h}^{(L-1)}, \boldsymbol{y}) = \mathbb{E}_{\boldsymbol{\Delta}\sim\mathcal{N}(0,\sigma^2\boldsymbol{I})}\left[\mathcal{L}\big((\boldsymbol{W}^{(L)}+\boldsymbol{\Delta})\boldsymbol{h}^{(L-1)}, \boldsymbol{y}\big)\right]. \tag{3}$$

This approach is related to approximating the softmax of a Gaussian distribution (Lu et al., 2021) and aligns with methods that focus on optimizing the last layer separately from the rest of the network (Newman et al., 2021). We can apply this smoothing to the cross-entropy loss and obtain the following result.

**Theorem 2** (Smoothed Cross-Entropy Loss)**.** *Let $\mathcal{L}_{\mathrm{CE}}$ be the cross-entropy loss, $\boldsymbol{h}^{(L-1)} \in \mathbb{R}^d$ be an input from the penultimate layer, $\boldsymbol{y} \in \mathbb{R}^c$ be a one-hot encoded label vector, and $\boldsymbol{W}^{(L)} \in \mathbb{R}^{c\times d}$ be a weight matrix. Then the following bound holds for the smoothed loss:*

$$\mathcal{L}^{\sigma}_{\mathrm{CE}}(\boldsymbol{W}^{(L)}\boldsymbol{h}^{(L-1)}, \boldsymbol{y}) \leq \mathcal{L}_{\mathrm{CE}}(\boldsymbol{W}^{(L)}\boldsymbol{h}^{(L-1)}, \boldsymbol{y}) + \frac{\sigma^2}{2}\|\boldsymbol{h}^{(L-1)}\|_2^2 . \tag{4}$$

Optimizing the right-hand side is equivalent to optimizing the original loss $\mathcal{L}_{\mathrm{CE}}$ with an added $\ell_2$ regularization term on the penultimate activations $\|\boldsymbol{h}^{(L-1)}\|_2^2$, this *activation decay* (AD) effect functions similarly to weight decay. As noise on inputs gives regularization on weights (Bishop, 1995), here noise on weights gives regularization on activations. Note that Taylor expansion gives a tighter approximation of the smoothed cross-entropy but the approximation is not an upper bound on the smoothed loss, see the result in Appendix.

## 4 EXPERIMENTS

### 4.1 THEORETICAL VALIDATION OF REGULARIZATION EFFECTS

This experiment empirically evaluates how closely the theoretical bound from Corollary 1 aligns with observed Hessian norms. Several runs of training on CIFAR-10 are performed with the ResNet-56 model and different parameters $\sigma$ for the AD. Then, we compute the largest eigenvalue of the Hessian on the final layers of the network to assess the sharpness reduction predicted by Gaussian

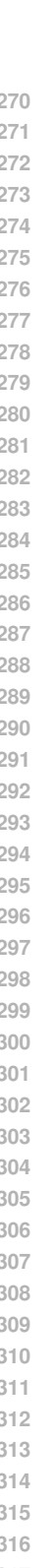
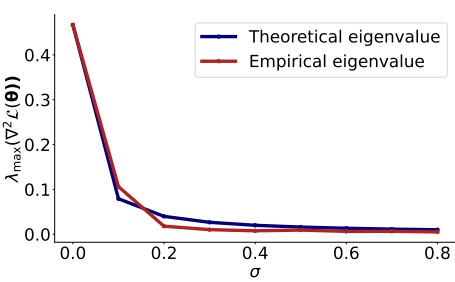
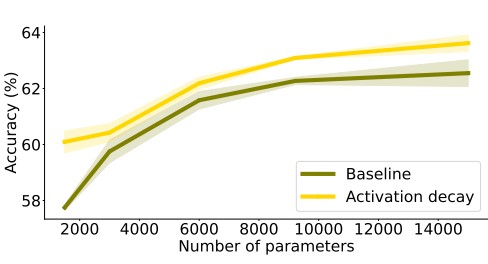

(a) Theoretical bound vs empirical Hessian.

(b) Accuracy vs number of features.

Figure 1: (a) Comparison between the theoretical bound on the largest eigenvalue of the Hessian given by Corollary 1 and the empirical value computed with PyHessian with a relative tolerance of $1e{-}3$, on a ResNet-56 model trained on CIFAR-10 for 300 epochs. (b) Accuracy on CIFAR-10 for an MLP with depth 3 and varying numbers of hidden features per layer. Our method with regularization is compared to the baseline with no regularization.

smoothing. Here Hessian is computed only w.r.t to $\mathbf{W}^{(\mathbf{L})}$ of the final layer. We estimate $\epsilon$ by averaging the gradient's norm near the minimum at the end of the training. The Hessian eigenvalue is computed on the training set at the end of the training after 300 epochs. We see in Figure 1a, that the theoretical bound gives the correct trend of the evolution of the curvature of the Hessian, the remaining mismatch might come from stochasticity in the Hessian eigenvalue computation, the Jensen gap, etc. We use the PyHessian library to compute the Hessian operator norm (Yao et al., 2020).

## 4.2 EMPIRICAL EVALUATION ON VISION DATASETS

**Classification with MLP on CIFAR-10** This experiment compares different regularization techniques on a 4-layer Multi-Layer Perceptron (MLP) network with GELU activation. The MLP has 3072 input features, and the training was conducted on the CIFAR-10 dataset. The model was trained using Stochastic Gradient Descent (SGD) without momentum, and no weight decay was applied. A learning rate scheduler with annealing was used to adjust the learning rate, which was set to $1e{-}1$. We use a batch size of 128 and standard data augmentation techniques, including random horizontal flips and random crops with padding of 4 pixels. These augmentations are applied to the training data to improve generalization. Each experiment was repeated 10 times to ensure statistical significance. We explored various regularization methods: We apply dropout (DO), parameterized by probability $p$, on intermediate layers $\boldsymbol{h}^{(l)}$; weight decay (WD) on all layers, parameterized by $\sigma$; activation decay (AD), parameterized by $\sigma$, on the last layer $\boldsymbol{h}^{(L)}$; a combination of AD on the last layer $\boldsymbol{h}^{(L)}$, parameterized by $\sigma$, with weight decay on intermediate layers $\boldsymbol{h}^{(l)}$, where the best parameter is obtained from the previous weight decay experiment; and SAM parametrized by $\rho$.

Results are presented in Figure 2. The baseline is at 62.17%. Our results demonstrate that AD increases generalization when $\sigma = 0.1$, improving accuracy by 2.63 %. Applying dropout alone also slightly enhances generalization, improving by 1.73% for best parameter $p = 0.1$. However, combining dropout with AD does not yield additional benefits and performs worse than using AD alone. Additionally, the application of intermediate loss regularization does not provide any noticeable benefits in terms of generalization. The method that combines AD and weight decay performs the best as highlighted by Theorem 1. SAM do not provide as good results for the MLP architecture as for CNNs architecture as reported in SAM paper (Foret et al., 2021). We also provide a comparison with a similar approach proposed by Baek et al. (2024) in Appendix A.1. Their method applies layer-wise activation regularization across all layers except the last, where weight decay is used. In contrast, our approach requires fewer hyperparameters to tune while achieving comparable results.

We also provide results on MLP-Mixer architecture (Tolstikhin et al., 2021) on ImageNet in the Appendix, showing that our method extends to a bigger dataset and architecture.

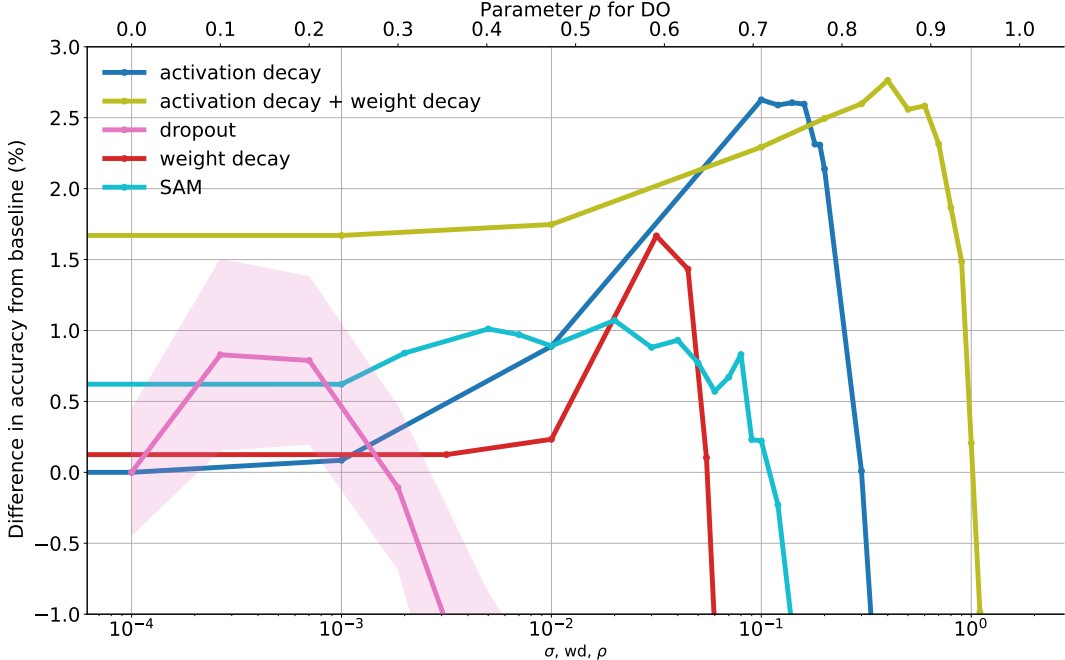

Figure 2: Comparison of accuracies of different regularizations, **the higher the better**, applied to a 4-layer MLP network trained on the CIFAR-10 dataset. The plots show the evolution of accuracy with varying values of parameter $\sigma$ for activation decay (AD) and weight decay (WD) and drop rate $p$ for dropout (DO). The first curve (AD) depicts the effect of $\sigma$ while keeping $p$ at $0.0$. The second curve (AD + WD) combines WD with the best parameter $1e-3$ and varying $\sigma$ for AD. The third curve (DO) illustrates the impact of dropout when $\sigma$ is $0.0$. The fourth curve (WD) depicts the effect of $\sigma$ when it parameterizes weight decay. The fourth curve (SAM) illustrates the impact of $\rho$ parameter. Shell indicates the standard deviation over 10 runs.

**Effect of overparameterization on regularization performance**   This experiment evaluates how AD regularization behaves when varying the number of parameters in the model. We use a 3-layer MLP on CIFAR-10 and adjust the number of hidden features to transition between underparameterized and overparameterized regimes. As shown in Figure 1b, we observe that the gains from our method are consistent across both regimes. Smoothing the loss leads to noticeable improvements in accuracy, demonstrating the effectiveness of our regularization technique regardless of the model's parameter count.

Table 1: Accuracies on validation set for baseline, AD ($\sigma = 0.2$), ASAM ($\rho_{\text{ASAM}} = 2.0$) and AD+SAM, for WideResNet on CIFAR-10. Results were reported after averaging over 3 runs and the standard deviation is $0.03$ for all runs.

| Configuration | Validation accuracy (%) |
| --- | --- |
| Baseline | 97.09 |
| ASAM | 97.48 |
| AD | 97.27 |
| AD + ASAM | **97.54** |

**Comparison to ASAM with Wide ResNet on CIFAR-10**   Table 1 presents a comparative analysis of the average accuracies on the validation set achieved by different training configurations using the WideResNet architecture trained on 300 epochs on the CIFAR-10 dataset. The configurations evaluated are: Baseline, the standard training setup without additional optimization techniques, achieved a validation accuracy of 97.09%. Ours, a proposed method utilizing AD with $\sigma = 0.2$, improved

the validation accuracy to 97.27%. For experiments involving SAM we use the upgrade Adaptive SAM (ASAM) (Kwon et al., 2021), we adopted the official implementation provided by the authors. To ensure a fair evaluation, we used the best parameter $\rho_{\text{ASAM}} = 2.0$ as specified in their code repository for this particular task which achieved a validation accuracy of 97.48%. AD + ASAM, a combination of the proposed method and SAM, yielded the highest validation accuracy of 97.54%. The best values of ASAM and AD parameters were picked by hyperparameter search. AD and ASAM outperform the baseline individually, while the combination yields the best performance. This can be explained because ASAM is designed to minimize the worst-case ascent sharpness, specifically targeting regions of the loss landscape where the maximum sharpness is reduced (Wen et al., 2022). In contrast, AD focuses on reducing the average sharpness, promoting a smoother and more stable optimization trajectory. This complementary behavior could explain why the two methods combine effectively, leading to enhanced generalization performance.

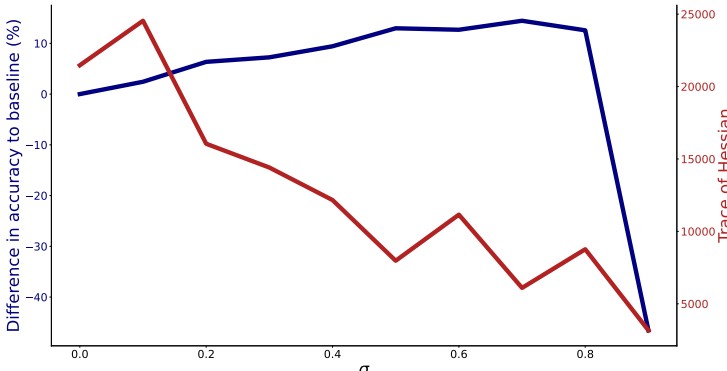

Figure 3: Accuracy difference and Hessian trace of the loss for varying $\sigma$, with 30% label noise on CIFAR-10 using ResNet-56.

**Classification with Label Noise with ResNet-56 on CIFAR-10**  We evaluate the effect of 30% label noise effect on CIFAR-10 classification using ResNet-56, varying the parameter $\sigma$ in AD. Figure 3 shows the accuracy difference from a baseline model and the trace of the Hessian of the loss w.r.t all parameters, plotted against $\sigma$. The Hessian trace, a proxy for sharpness, indicates sharper minima and poorer generalization under label noise. As $\sigma$ increases, accuracy initially improves but declines as the Hessian trace rises, aligning with findings from Foret et al. (2021) and Baek et al. (2024), where SAM (Sharpness-Aware Minimization) improves regularization under label noise. Our AD method enhances noise robustness by controlling sharpness as $\sigma$ increases, improving accuracy without the computational cost of SAM. This makes AD an efficient alternative for handling noisy labels for free. This experiment further reinforced the link between flat minima and label noise robustness.

## 4.3 ADAPTING TO LARGE LANGUAGE MODEL

Multi-task learning (MTL) Zhang et al. (2021) is a paradigm in machine learning where a model is trained to perform multiple tasks simultaneously, leveraging shared representations across tasks. MTL also offers the benefit of reducing computational costs and latency, enabling the model to handle multiple tasks in a single inference step, rather than performing separate inferences for each task.

With the advent of large-scale pretrained models, multi-task learning has become increasingly popular in NLP. Pretrained on large corpora, these models can capture a wide range of patterns and dependencies in text data. Leveraging this general knowledge during fine-tuning while specializing in specific tasks leads to new state-of-the-art performances on a variety of downstream tasks. This task-specific adaptation during fine-tuning, is needed for earlier Large Language Models (LLMs) as BERT Devlin et al. (2019) and RoBERTa Liu et al. (2019) and has recently been reduced to few-shot learning with models like GPT-3 Brown et al. (2020) and T5 Raffel et al. (2020) and even extended to zero-shot settings by Wei et al. (2022); Sanh et al. (2022) in the context of very large models.

However, fine-tuning multiple tasks can lead to overfitting on individual tasks, which can degrade the model's performance on other tasks, calling for specific design choices, adding extra task-specific parameters (adapters), or using specific prompts (see Stickland & Murray (2019); Wang et al. (2023)). Our AD method promotes flat minima to improve generalization across tasks. The goal is to prevent overfitting to any specific task while retaining the benefits of pretraining on various tasks. Our flat minima regularization helps preserve the generalization capabilities of the pretrained model, ensuring robust performance across all tasks during fine-tuning. As detailed in Section 2.1 flat minima and generalization are often tied together and the experimental results presented in Tables 6 and 3 show our regularization helps conserve performance across diverse tasks by promoting smooth optimization landscapes during fine-tuning.

**Fine tuning on distinct tasks**  In this experiment, we evaluate the performance of a multi-task NLP model using the RoBERTa (Liu et al., 2019), BERT (Devlin et al., 2019), and T5 (Raffel et al., 2020) architectures to handle distinct tasks. The multi-task setup allows the model to process these tasks simultaneously, optimizing latency and resource consumption. We experimentally show that in the context of fine-tuning and few-shot learning, our Activation Decay (AD) method helps LLMs to generalize better across tasks, leading to improved performance compared to the baseline models.

Each task is described as follows: (i) Sentiment Analysis: a binary classification task (positive/negative) on the IMDb dataset, using classification accuracy as the metric; (ii) NER: named entity recognition on the Snips and CoNLL datasets, evaluated using F1-score, precision, and recall metrics; (iii) Intent Classification: intent detection on the Snips dataset, evaluated with classification accuracy; (iv) Entailment Classification (SNLI): a binary classification task predicting whether a sentence entails another, based on the SNLI dataset, with classification accuracy as the metric; (v) POS Tagging: part-of-speech tagging on the CoNLL dataset, evaluated using F1-score, precision, and recall metrics; (vi) Query Correctness: a binary classification task assessing the correctness of queries, evaluated with classification accuracy.

The backbone models used for all tasks are BERT (bert-base) and RoBERTa (roberta-base), with dropout probability set to 0. The model's performance is evaluated using a custom smoothed loss function with $\sigma = 0.05$, and the results are compared with SAM regularization at different $\rho$ values. Weight decay is present by default in the training configuration of the backbone. We use standard training configuration from HuggingFace corresponding models and trainers.

Table 2 presents evaluation results for fine-tuning BERT on seven NLP tasks, comparing the baseline model with standard dropout ($p = 0.1$), SAM regularization with standard values ($\rho = 0.01$), and Activation Decay ($\sigma = 0.05$). The best values for SAM were selected from $\rho = 0.01, 0.05, 0.1$, and Activation Decay from $\sigma = 0.01, 0.05, 0.1$. Metrics include classification accuracy, F1-score, precision, and recall. Activation Decay consistently outperforms both the baseline and SAM across most tasks. The same results for RoBERTa are reported in Table 5 in the Appendix, showing the efficiency of AD.

**Few shots learning on Multilingual Massive Multitask Language Understanding (MMMLU) dataset**  We evaluate our models on the MMMLU dataset (Hendrycks et al., 2020), which spans 57 diverse topics ranging from elementary to advanced professional subjects. We use the newly updated version published by (OpenAI, 2023), which expanded the dataset to include 14 languages using professional human translators. We use the T5 architecture (Raffel et al., 2020) for our experiments, fine-tuning the small (60 M), base (220 M), and large (770 M) variants of the model over 3 epochs. The results, summarized in Table 3, show that our AD with $\sigma = 0.01$ in a multitask setting, consistently outperforms the standard dropout $p = 0.1$ a common baseline for fine-tuning baseline across all model sizes, demonstrating its effectiveness for large models. This result highlights the importance of our approach in improving accuracy, particularly in large-scale multitask environments. All code and implementation details will be made available upon acceptance of the paper to ensure reproducibility.

## 5 DISCUSSION AND CONCLUSION

The formula in Theorem 1 offers guidance for layer-specific regularization by applying either weight decay or activation decay. Since most of the training configurations already employed weight decay,

Table 2: Evaluation results for BERT baseline with DO ($p = 0.1$), SAM ($\rho = 0.01$), and AD ($\sigma = 0.05$) on 7 tasks.

| Metric | DO | SAM | AD |
|---|---|---|---|
| **Sentiment Evaluation** | | | |
| Classification Accuracy (%) | 76.72 | 76.54 | **77.08** |
| **NER Evaluation** | | | |
| Snips F1 Score (%) | 78.33 | 69.67 | **80.90** |
| Snips Precision (%) | 73.69 | 64.11 | **76.20** |
| Snips Recall (%) | 83.59 | 76.28 | **86.21** |
| **Intent Evaluation** | | | |
| Classification Accuracy (%) | 98.04 | 98.19 | **98.49** |
| **Entailment SNLI Evaluation** | | | |
| Classification Accuracy (%) | 87.96 | **89.39** | 88.88 |
| **CoNLL NER Evaluation** | | | |
| Seqeval F1 Score (%) | 64.43 | 61.01 | **65.94** |
| Seqeval Precision (%) | 61.87 | 61.48 | **64.11** |
| Seqeval Recall (%) | 67.20 | 60.55 | **67.87** |
| **CoNLL POS Evaluation** | | | |
| Seqeval F1 Score (%) | 75.95 | 72.48 | **77.89** |
| Seqeval Precision (%) | 74.89 | 71.59 | **76.98** |
| Seqeval Recall (%) | 77.04 | 73.39 | **78.82** |
| **Query Correctness Evaluation** | | | |
| Classification Accuracy (%) | **69.95** | 69.47 | 69.31 |

Table 3: Test accuracy results for T5 configurations on the MMMLU dataset, for baseline used with DO ($p = 0.1$), and AD ($\sigma = 0.01$).

| Model | DO | AD |
|---|---|---|
| T5-large | 52.07 | **52.95** |
| T5-base | 49.89 | **50.25** |
| T5-small | 32.21 | **33.49** |

we introduced activation decay to replace dropout, as the latter is a noise-based injection method. Empirical results confirm the effectiveness of this replacement and validate the proposed formula. We also experimented with proportional weight decay across layers, aiming to scale it based on each layer's contribution to the overall Hessian spectral norm of the loss. However, this approach did not yield improved results, potentially due to the looseness of the formula or the fact that weight decay controls the Frobenius norm, which serves as a loose upper bound on the spectral norm. Future work could focus on refining the formula to more tightly align with the spectral norm improve proportional weight decay performance and study the impact of normalization layers like Batch Normalization and Layer Normalization on the Hessian spectral norm.

The proposed method of activation decay presents a novel and effective approach to improving generalization in deep learning by addressing the sharpness of minima during training. By leveraging Gaussian smoothing to regularize critical activations, activation decay flattens minima and enhances robustness without incurring additional computational costs. Our method provides a deterministic alternative to stochastic regularization techniques like dropout and SAM, maintaining efficiency while achieving comparable or superior performance. Additionally, its ability to integrate and combine seamlessly with existing regularization methods like weight decay makes it a versatile and practical tool for large-scale models.

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

## A    ADDITIONAL EXPERIMENTS

### A.1    CLASSIFICATION WITH MLP ON CIFAR-10

In this experiment we use the same setting as in experiemnt 4.2. We conducted a comparison of our method AD, with the approach outlined in (Baek et al., 2024) Section 4.3, whose framework, designed specifically for label noise robustness, requires tuning decay parameters for each layer to achieve effective regularization. This introduces additional hyperparameter complexity but shares conceptual similarities with our approach.

To ensure a fair comparison, both methods were evaluated under identical experimental settings, including hyperparameter tuning for  Baek et al. (2024)'s regularization coefficients. The results are summarized in Table 4.

Table 4: Comparison with (Baek et al., 2024) and Activation Decay (AD) under identical experimental settings.

| Metric | (Baek et al., 2024) | AD (ours) |
|---|---|---|
| Mean test accuracy (%) | 65.05 | 65.11 |
| 95% confidence interval | [64.86, 65.23] | [64.90, 65.33] |

Our findings demonstrate that AD, which applies $\ell_2$-regularization to the penultimate activations, achieves comparable results to (Baek et al., 2024)'s method without the need for layer-wise tuning. Specifically,  (Baek et al., 2024)'s method requires tuning separate coefficients for intermediate layers ($\sigma = 1e-3$) and for the last layer parameters ($\sigma = 1e-2$). In contrast, our approach eliminates this complexity and achieves similar performance with a single hyperparameter, $\sigma$, set to $0.1$.

The simplicity of our method reduces the number of hyperparameters to tune, making it both more practical and easier to analyze. Furthermore, regularizing the penultimate activation indirectly regularizes preceding layers, as the penultimate activation encapsulates their contributions.

### A.2    CLASSIFICATION WITH MLP-MIXER ON IMAGENET-1K

We test our method on the MLP-Mixer architecture  (Tolstikhin et al., 2021), using the same settings as Liu et al. (2023) and Liu et al. (2022). We applied AD to the final layer and the cross-entropy loss of the Mixer-S/32 architecture on ImageNet-1k and reported the accuracy. Figure 4 results indicate that AD improves model performance, leading to higher accuracy than the baseline.

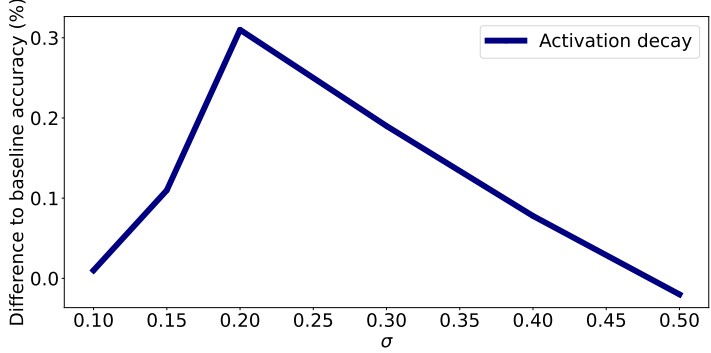

Figure 4: Plot of different pieces of training on ImageNet of MLP-Mixer with AD, with varying $\sigma$.

### A.3    EXPERIMENTS ON LLM

Table 5: Evaluation results for RoBERTa baseline and AD ($\sigma = 0.05$), on 7 tasks.

| Metric | DO | AD |
|---|---|---|
| **Sentiment Evaluation** | | |
| Classification Accuracy (%) | 77.66 | **77.68** |
| **NER Evaluation** | | |
| Snips F1 Score (%) | 72.70 | **73.99** |
| Snips Precision (%) | 67.78 | **69.25** |
| Snips Recall (%) | 78.39 | **79.44** |
| **Intent Evaluation** | | |
| Classification Accuracy (%) | **97.59** | 96.83 |
| **Entailment SNLI Evaluation** | | |
| Classification Accuracy (%) | 88.60 | **89.33** |
| **CoNLL NER Evaluation** | | |
| Seqeval F1 Score (%) | 67.62 | **67.90** |
| Seqeval Precision (%) | 65.08 | **65.38** |
| Seqeval Recall (%) | 70.36 | **70.61** |
| **CoNLL POS Evaluation** | | |
| Seqeval F1 Score (%) | 71.86 | **72.14** |
| Seqeval Precision (%) | 70.68 | **70.89** |
| Seqeval Recall (%) | 73.07 | **73.45** |
| **Query Correctness Evaluation** | | |
| Classification Accuracy (%) | **68.08** | 68.06 |

Table 6: Evaluation results for BERT baseline with DO ($p = 0.1$), SAM for different $\rho$ values, and AD with ($\sigma = 0.05$) on 7 tasks.

| Metric | DO | SAM | | | AD |
|---|---|---|---|---|---|
| | | $\rho = 0.01$ | $\rho = 0.05$ | $\rho = 0.1$ | |
| **Sentiment Evaluation** | | | | | |
| Classification Accuracy (%) | 76.72 | 76.54 | 75.38 | 62.28 | **77.08** |
| **NER Evaluation** | | | | | |
| Snips F1 Score (%) | 78.33 | 69.67 | 62.29 | 63.95 | **80.90** |
| Snips Precision (%) | 73.69 | 64.11 | 56.58 | 58.08 | **76.20** |
| Snips Recall (%) | 83.59 | 76.28 | 69.27 | 71.14 | **86.21** |
| **Intent Evaluation** | | | | | |
| Classification Accuracy (%) | 98.04 | 98.19 | 97.43 | 97.28 | **98.49** |
| **Entailment SNLI Evaluation** | | | | | |
| Classification Accuracy (%) | 87.96 | **89.39** | 86.77 | 83.85 | 88.88 |
| **CoNLL NER Evaluation** | | | | | |
| Seqeval F1 Score (%) | 64.43 | 61.01 | 51.09 | 52.05 | **65.94** |
| Seqeval Precision (%) | 61.87 | 61.48 | 51.64 | 51.72 | **64.11** |
| Seqeval Recall (%) | 67.20 | 60.55 | 50.56 | 52.39 | **67.87** |
| **CoNLL POS Evaluation** | | | | | |
| Seqeval F1 Score (%) | 75.95 | 72.48 | 69.31 | 71.20 | **77.89** |
| Seqeval Precision (%) | 74.89 | 71.59 | 68.79 | 70.46 | **76.98** |
| Seqeval Recall (%) | 77.04 | 73.39 | 69.84 | 71.97 | **78.82** |
| **Query Correctness Evaluation** | | | | | |
| Classification Accuracy (%) | **69.95** | 69.47 | 66.24 | 64.74 | 69.31 |

# B PROOFS

## B.1 PROOF OF COROLLARY 1

*Proof.* Proof is the same as the one from Theorem 2 of Delattre et al. (2024) but one has to simply adapt the bound with $\|\nabla\mathcal{L}(\boldsymbol{\theta})\|_2 \leq \epsilon$ instead of having $\nabla\mathcal{L}(\boldsymbol{\theta})$ in the simplex. $\qquad\square$

## B.2 PROOF OF THEOREM 1

*Proof.* We aim to derive an upper bound on the operator norm of the Hessian matrix $\nabla_{\boldsymbol{\theta}}^2 \mathcal{L}(\boldsymbol{\theta})$ at a point where the loss function attains zero: $\mathcal{L}(\boldsymbol{\theta}) = 0$.

Let us denote $\mathbf{z} = \mathbf{h}^{(L-1)}$ as the penultimate activation.

The Hessian matrix of $\mathcal{L}$ with respect to $\boldsymbol{\theta}$ is given by:

$$\nabla_{\boldsymbol{\theta}}^2 \mathcal{L}(\boldsymbol{\theta}) = \frac{\partial^2 \mathcal{L}}{\partial \boldsymbol{\theta}^2} \, ,$$

and using the chain rule:

$$\nabla_{\boldsymbol{\theta}}^2 \mathcal{L} = \left(\frac{\partial^2 \mathbf{z}}{\partial \boldsymbol{\theta}^2}\right)^\top \nabla_{\mathbf{z}}\mathcal{L} + \left(\frac{\partial \mathbf{z}}{\partial \boldsymbol{\theta}}\right)^\top \nabla_{\mathbf{z}}^2 \mathcal{L}\left(\frac{\partial \mathbf{z}}{\partial \boldsymbol{\theta}}\right) \, .$$

Since $\mathcal{L}(\boldsymbol{\theta}) = 0$, this implies that $\boldsymbol{\theta}$ is at a global minimum of $\mathcal{L}$ due to the non-negativity of the loss function. Therefore, by the necessary condition for optimality in differentiable functions, the gradient of the loss function with respect to $\boldsymbol{\theta}$ vanishes:

$$\nabla\mathcal{L}(\boldsymbol{\theta}) = 0.$$

$$\nabla_{\boldsymbol{\theta}}^2 \mathcal{L} = \left(\frac{\partial \mathbf{z}}{\partial \boldsymbol{\theta}}\right)^\top \nabla_{\mathbf{z}}^2 \mathcal{L}\left(\frac{\partial \mathbf{z}}{\partial \boldsymbol{\theta}}\right) \, .$$

We expand $\partial\mathbf{z}/\partial\boldsymbol{\theta}$ using the chain rule:

$$\frac{\partial \mathbf{z}}{\partial \boldsymbol{\theta}} = \sum_{j=1}^{L-1} \frac{\partial \mathbf{z}}{\partial \mathbf{h}^{(j)}} \frac{\partial \mathbf{h}^{(j)}}{\partial \boldsymbol{\theta}} \, ,$$

and each term $\partial\mathbf{z}/\partial\mathbf{h}^{(j)}$ involves the product of derivatives from layer $j+1$ to $L-1$:

$$\frac{\partial \mathbf{z}}{\partial \mathbf{h}^{(j)}} = \prod_{l=j+1}^{L-1} \frac{\partial f^{(l)}}{\partial \mathbf{h}^{(l-1)}} \, .$$

Thus, the Jacobian becomes:

$$\frac{\partial \mathbf{z}}{\partial \boldsymbol{\theta}} = \sum_{j=1}^{L-1} \left(\prod_{l=j+1}^{L-1} \frac{\partial f^{(l)}}{\partial \mathbf{h}^{(l-1)}}\right) \frac{\partial \mathbf{h}^{(j)}}{\partial \boldsymbol{\theta}} \, .$$

Applying the triangle inequality for operator norms:

$$\left\|\frac{\partial \mathbf{z}}{\partial \boldsymbol{\theta}}\right\|_2 \leq \sum_{j=1}^{L-1} \left\|\prod_{l=j+1}^{L-1} \frac{\partial f^{(l)}}{\partial \mathbf{h}^{(l-1)}}\right\|_2 \left\|\frac{\partial \mathbf{h}^{(j)}}{\partial \boldsymbol{\theta}}\right\|_2$$

For contractant non-linear activation such as ReLU and GELU, the layer's derivative is bounded by its weight spectral norm:

$$\left\|\frac{\partial f^{(l)}}{\partial \mathbf{h}^{(l-1)}}\right\|_2 \leq \left\|\boldsymbol{W}^{(l)}\right\|_2$$

we further bound the product of norms:

$$\left\| \prod_{l=j+1}^{L-1} \frac{\partial f^{(l)}}{\partial \mathbf{h}^{(l-1)}} \right\|_2 \leq \prod_{l=j+1}^{L-1} \left\| \boldsymbol{W}^{(l)} \right\|_2$$

Thus, the operator norm of the Jacobian is bounded by:

$$\left\| \frac{\partial \mathbf{z}}{\partial \boldsymbol{\theta}} \right\|_2 \leq \sum_{j=1}^{L-1} \left\| \frac{\partial \mathbf{h}^{(j)}}{\partial \boldsymbol{\theta}} \right\|_2 \prod_{l=j+1}^{L-1} \left\| \boldsymbol{W}^{(l)} \right\|_2$$

Since $\mathbf{z} = \mathbf{h}^{(L-1)}$, we can write:

$$\| \nabla_{\boldsymbol{\theta}}^2 \, \mathcal{L}(\boldsymbol{\theta}) \|_2 \leq \left( \sum_{j=1}^{L-1} \left\| \frac{\partial \mathbf{h}^{(j)}}{\partial \boldsymbol{\theta}} \right\|_2 \prod_{l=j+1}^{L-1} \left\| \boldsymbol{W}^{(l)} \right\|_2 \right)^2 \left\| \nabla_{\mathbf{h}^{(L-1)}}^2 \mathcal{L}(\boldsymbol{\theta}) \right\|_2 \ .$$

$\square$

### B.3 PROOF OF THEOREM 2

*Proof.* The original cross-entropy loss for the correct class $c$ is given by:

$$\mathcal{L}(h_L(\boldsymbol{h}_{L-1}), \boldsymbol{y}) = -\boldsymbol{W}_c^\top \boldsymbol{h}_{L-1} + \log \left( \sum_{j=1}^{d} \exp(\boldsymbol{W}_j^\top \boldsymbol{h}_{L-1}) \right) \ .$$

Consider the smoothed loss by introducing Gaussian noise $\boldsymbol{\Delta} \sim \mathcal{N}(0, \boldsymbol{I}\sigma^2)$:

$$\mathcal{L}^\sigma(h_L(\boldsymbol{h}_{L-1}), \boldsymbol{y}) = \mathbb{E}_{\boldsymbol{\Delta} \sim \mathcal{N}(0, \boldsymbol{I}\sigma^2)} \left[ -(\boldsymbol{W}_c + \boldsymbol{\Delta}_c)^\top \boldsymbol{h}_{L-1} + \log \left( \sum_{j=1}^{d} \exp((\boldsymbol{W}_j + \boldsymbol{\Delta}_j)^\top \boldsymbol{h}_{L-1}) \right) \right]$$

Separating the terms:

$$\mathcal{L}^\sigma(h_L(\boldsymbol{h}_{L-1}), \boldsymbol{y}) = -\boldsymbol{W}_c^\top \boldsymbol{h}_{L-1} + \mathbb{E}_{\boldsymbol{\Delta} \sim \mathcal{N}(0, \boldsymbol{I}\sigma^2)} \left[ \log \left( \sum_{j=1}^{d} \exp((\boldsymbol{W}_j + \boldsymbol{\Delta}_j)^\top \boldsymbol{h}_{L-1}) \right) \right] \ .$$

By applying Jensen's inequality on the expectation inside the logarithm, we get:

$$\mathbb{E}_{\boldsymbol{\Delta}} \left[ \log \left( \sum_{j=1}^{d} \exp((\boldsymbol{W}_j + \boldsymbol{\Delta}_j)^\top \boldsymbol{h}_{L-1}) \right) \right] \leq \log \left( \mathbb{E}_{\boldsymbol{\Delta}} \left[ \sum_{j=1}^{d} \exp((\boldsymbol{W}_j + \boldsymbol{\Delta}_j)^\top \boldsymbol{h}_{L-1}) \right] \right) \ .$$

Since $\boldsymbol{\Delta}_j \sim \mathcal{N}(0, \sigma^2 \boldsymbol{I})$, we use the moment generating function of the Gaussian distribution:

$$\mathbb{E}[e^Z] = e^{\mu + \frac{1}{2}\sigma^2}$$

Applying this to our case for each $\boldsymbol{W}_j^\top \boldsymbol{h}_{L-1} + \boldsymbol{\Delta}_j^\top \boldsymbol{h}_{L-1}$:

$$\mathbb{E}\left[ \exp((\boldsymbol{W}_j + \boldsymbol{\Delta}_j)^\top \boldsymbol{h}_{L-1}) \right] = \exp(\boldsymbol{W}_j^\top \boldsymbol{h}_{L-1}) \mathbb{E}\left[ \exp(\boldsymbol{\Delta}_j^\top \boldsymbol{h}_{L-1}) \right]$$

Given $\boldsymbol{\Delta}_j \sim \mathcal{N}(0, \sigma^2 \boldsymbol{I})$ and $\boldsymbol{\Delta}_j^\top \boldsymbol{h}_{L-1}$ is a Gaussian with mean 0 and variance $\sigma^2 \|\boldsymbol{h}_{L-1}\|^2$, we get:

$$\mathbb{E}\left[ \exp(\boldsymbol{\Delta}_j^\top \boldsymbol{h}_{L-1}) \right] = \exp \left( 0 + \frac{1}{2}\sigma^2 \|\boldsymbol{h}_{L-1}\|^2 \right)$$

Therefore,

$$\mathbb{E}\left[ \exp((\boldsymbol{W}_j + \boldsymbol{\Delta}_j)^\top \boldsymbol{h}_{L-1}) \right] = \exp \left( \boldsymbol{W}_j^\top \boldsymbol{h}_{L-1} + \frac{1}{2}\sigma^2 \|\boldsymbol{h}_{L-1}\|^2 \right)$$

Summing over $j$:

$$\mathbb{E}_{\boldsymbol{\Delta}}\left[\sum_{j=1}^{d}\exp((\boldsymbol{W}_j+\boldsymbol{\Delta}_j)^{\top}\boldsymbol{h}_{L-1})\right]=\sum_{j=1}^{d}\exp\left(\boldsymbol{W}_j^{\top}\boldsymbol{h}_{L-1}+\frac{1}{2}\sigma^2\|\boldsymbol{h}_{L-1}\|^2\right)$$

Substituting this back into the expression for the smoothed loss, we obtain:

$$\mathcal{L}^{\sigma}(h_L(\boldsymbol{h}_{L-1}),\boldsymbol{y})\leq-\boldsymbol{W}_c^{\top}\boldsymbol{h}_{L-1}+\log\left(\sum_{j=1}^{d}\exp\left(\boldsymbol{W}_j^{\top}\boldsymbol{h}_{L-1}+\frac{1}{2}\sigma^2\|\boldsymbol{h}_{L-1}\|^2\right)\right)$$

This result shows that the smoothed loss $\mathcal{L}^{\sigma}(h_L(\boldsymbol{h}_{L-1}),\boldsymbol{y})$ is bounded above by the original loss with an additional offset term $\frac{1}{2}\sigma^2\|\boldsymbol{h}_{L-1}\|^2$. This offset is akin to the regularization term observed in the loss from the work of Tsuzuku et al. (2018) in term of $L\epsilon$ where $L$ is the Lipschitz constant and $\epsilon$ the size of the perturbation. $\qquad\square$

### B.4 THEOREM AND PROOF ON TIGHTER APPROXIMATION USING TAYLOR EXPANSION

**Theorem 3** (Tighter Approximation via Taylor Expansion). *Let $\boldsymbol{W}^{(L)}\in\mathbb{R}^{c\times d}$, $\boldsymbol{h}^{(L-1)}\in\mathbb{R}^d$, and $\boldsymbol{\Delta}\in\mathbb{R}^{c\times d}$ with elements drawn independently from $\mathcal{N}(0,\sigma^2)$. Denote $\hat{\boldsymbol{y}}=\text{softmax}(\boldsymbol{W}^{(L)}\boldsymbol{h}^{(L-1)})$. For small $\sigma$, the expected cross-entropy loss under perturbations $\boldsymbol{\Delta}$ is approximately:*

$$\mathbb{E}\left[\mathcal{L}_{\text{CE}}\big((\boldsymbol{W}^{(L)}+\boldsymbol{\Delta})\boldsymbol{h}^{(L-1)},\boldsymbol{y}\big)\right]\approx\mathcal{L}_{\text{CE}}(\boldsymbol{W}^{(L)}\boldsymbol{h}^{(L-1)},\boldsymbol{y})+\tfrac{1}{2}\sigma^2\|\boldsymbol{h}^{(L-1)}\|_2^2\sum_{i=1}^{c}\hat{y}_i(1-\hat{y}_i)\,.$$

Note that the obtained approximation of the smoothed loss is not an upper bound on the exact smoothed loss.

*Proof.* We start by expanding the cross-entropy loss around $\boldsymbol{W}^{(L)}\boldsymbol{h}^{(L-1)}$ using a first-order Taylor expansion:

$$\mathcal{L}_{\text{CE}}\big((\boldsymbol{W}^{(L)}+\boldsymbol{\Delta})\boldsymbol{h}^{(L-1)},\boldsymbol{y}\big)\approx\mathcal{L}_{\text{CE}}(\boldsymbol{W}^{(L)}\boldsymbol{h}^{(L-1)},\boldsymbol{y})+\nabla\mathcal{L}_{\text{CE}}(\boldsymbol{W}^{(L)}\boldsymbol{h}^{(L-1)},\boldsymbol{y})^{\top}(\boldsymbol{\Delta}\boldsymbol{h}^{(L-1)})\,,$$

where $\nabla\mathcal{L}_{\text{CE}}(\boldsymbol{W}^{(L)}\boldsymbol{h}^{(L-1)},\boldsymbol{y})=\hat{\boldsymbol{y}}-\boldsymbol{y}$. The first-order term is then:

$$(\hat{\boldsymbol{y}}-\boldsymbol{y})^{\top}\boldsymbol{\Delta}\boldsymbol{h}^{(L-1)}\,.$$

Taking the expectation of this term with respect to $\boldsymbol{\Delta}$, we use the fact that $\mathbb{E}[\boldsymbol{\Delta}]=0$, so the expectation of the first-order term is zero:

$$\mathbb{E}\left[(\hat{\boldsymbol{y}}-\boldsymbol{y})^{\top}\boldsymbol{\Delta}\boldsymbol{h}^{(L-1)}\right]=0\,.$$

We then proceed with the second-order Taylor expansion:

$$\frac{1}{2}(\boldsymbol{\Delta}\boldsymbol{h}^{(L-1)})^{\top}\nabla^2\mathcal{L}_{\text{CE}}(\boldsymbol{W}^{(L)}\boldsymbol{h}^{(L-1)},\boldsymbol{y})(\boldsymbol{\Delta}\boldsymbol{h}^{(L-1)})\,,$$

where the Hessian $\nabla^2\mathcal{L}_{\text{CE}}(\boldsymbol{W}^{(L)}\boldsymbol{h}^{(L-1)},\boldsymbol{y})$ is given by:

$$\nabla^2\mathcal{L}_{\text{CE}}(\boldsymbol{W}^{(L)}\boldsymbol{h}^{(L-1)},\boldsymbol{y})=\text{diag}(\hat{\boldsymbol{y}})-\hat{\boldsymbol{y}}\hat{\boldsymbol{y}}^{\top}\,.$$

Now, we compute the expectation of the second-order term. Using the property of quadratic forms for Gaussian random variables, we have:

$$\mathbb{E}[\boldsymbol{\Delta}\boldsymbol{h}^{(L-1)}(\boldsymbol{\Delta}\boldsymbol{h}^{(L-1)})^{\top}]=\sigma^2\|\boldsymbol{h}^{(L-1)}\|_2^2\boldsymbol{I}_c\,,$$

where $\boldsymbol{I}_c$ is the identity matrix in $\mathbb{R}^{c\times c}$. Thus, the second-order term simplifies to:

$$\frac{\sigma^2}{2}\|\boldsymbol{h}^{(L-1)}\|_2^2\text{tr}\left(\nabla^2\mathcal{L}_{\text{CE}}(\boldsymbol{W}^{(L)}\boldsymbol{h}^{(L-1)},\boldsymbol{y})\right)\,.$$

Finally, we compute the trace of the Hessian:

$$\text{tr}\left(\text{diag}(\hat{\boldsymbol{y}}) - \hat{\boldsymbol{y}}\hat{\boldsymbol{y}}^\top\right) = \sum_{i=1}^{c} \hat{y}_i(1 - \hat{y}_i),$$

as the trace of $\hat{\boldsymbol{y}}\hat{\boldsymbol{y}}^\top$ is 1. Therefore, the second-order term becomes:

$$\frac{\sigma^2}{2}\|\boldsymbol{h}^{(L-1)}\|_2^2 \sum_{i=1}^{c} \hat{y}_i(1 - \hat{y}_i).$$

Thus, the total approximation including both the first- and second-order terms is:

$$\mathbb{E}\left[\mathcal{L}_{\text{CE}}\left((\boldsymbol{W}^{(L)} + \boldsymbol{\Delta})\boldsymbol{h}^{(L-1)}, \boldsymbol{y}\right)\right] \approx \mathcal{L}_{\text{CE}}(\boldsymbol{W}^{(L)}\boldsymbol{h}^{(L-1)}, \boldsymbol{y}) + \frac{\sigma^2}{2}\|\boldsymbol{h}^{(L-1)}\|_2^2 \sum_{i=1}^{c} \hat{y}_i(1 - \hat{y}_i).$$

$\square$

