# OpenReview forum: "Activation Decay by Loss Smoothing to Enhance Generalization"
_ICLR.cc/2025/Conference — ICLR 2025 Conference Withdrawn Submission_

### Official Review · Reviewer_3SWC · 2024-10-31

**Soundness:** 2
**Presentation:** 2
**Contribution:** 2
**Rating:** 3
**Confidence:** 4

**Summary:**

The authors introduce a method called as "Activation Decay", which regularizes the activation in the penultimate layer to flatten sharp minima and improve generalization across a range of tasks. This approach is grounded in a theoretical framework that connects noise variance to the spectral norm of the Hessian, demonstrating that the activation in the penultimate layer serves as an upper bound for Gaussian smoothing on Hessian curvature. There are experiments on CIFAR-10, ImageNet, and NLP benchmarks.

**Strengths:**

- The paper is easy to follow.
- The idea of regularizing only the activation in the penultimate layer is quite interesting, and it can replace dropout at this layer.

**Weaknesses:**

- The idea of regularizing later layers has been explored in Baek et al. [2024]. This effectively reduces SAM's effectiveness to that of SGD with an L2 norm penalty on the intermediate activations and the weights of the last layer in a two-layer, deep linear network. Could you compare your method with theirs?
- Activation decay can only be applied to the penultimate layer, whereas dropout can be applied in multiple locations within the backbone architecture. Therefore, the use of activation decay might be limited.
- Figure 1a lacks annotations for better clarity.
- In Figure 1a, why is the estimated Hessian calculated only with respect to the final layer? Does this mismatch the theoretical results in Theorems 1 and 2?
- In Table 1, this paper compares its methods with ASAM [Kwon et al., 2021]  but is missing a citation for it. Why don’t the authors compare their methods with the original SAM?

References:
- Christina Baek, Zico Kolter, and Aditi Raghunathan. Why is sam robust to label noise?, 2024. URL https://arxiv.org/abs/2405.03676.
- Jungmin Kwon, Jeongseop Kim, Hyunseo Park, and In Kwon Choi. Asam: Adaptive sharpness-aware minimization for scale-invariant learning of deep neural networks. In International Conference on Machine Learning, pages 5905–5914. PMLR, 2021.

**Questions:**

- In Table 2, could you explain why the optimal perturbation for your experiments with SAM is lower ($\rho = 0.01$) compared to the optimal $\rho = 0.15$ recommended for NLP tasks in Bahri et al. [2022]?
- The improvement of proposed methods in Section 4.2 is not significant. Could you extend your experiments to more complex datasets, such as CIFAR-100 and Tiny ImageNet, so that the differences and improvements are clearer?
- See the Weaknesses part.

References:
- Dara Bahri, Hossein Mobahi, and Yi Tay. Sharpness-aware minimization improves language model generalization, 2022. URL https://arxiv.org/abs/2110.08529.

---

> ### Author Response · Authors · 2024-11-18
>
> We thank the reviewer for their precious feedbacks, please find a revised version of the paper uploaded.
>
> # Weakness
> ## W1)
> "The idea of regularizing later layers has been explored in Baek et al. [2024]. This effectively reduces SAM's effectiveness to that of SGD with an L2 norm penalty on the intermediate activations and the weights of the last layer in a two-layer, deep linear network. Could you compare your method with theirs?"
>
> We appreciate the suggestion to compare our method with Baek et al. [2024]. Their framework, designed specifically for label noise robustness, involves tuning decay parameters for each layer to achieve effective regularization. This requirement introduces additional hyperparameter complexity but also shares conceptual similarities with our approach.
>
> To address the reviewer's suggestion, we conducted experiments to directly compare our method, **Activation Decay (AD)**, with the approach outlined in Baek et al. Both methods were evaluated under identical experimental settings for a fair comparison, including hyperparameter tuning for Baek et al.'s regularization coefficients.
>
> The results are summarized in the table below:
>
> | Metric                  | **Baek et al.**      | **AD (ours) ($\sigma=0.1)$**         |
> |-------------------------|------------------|-------------------|
> | Mean Test Accuracy (%)  | 65.05            | 65.11             |
> | 95% Confidence Interval | [64.86, 65.23]   | [64.90, 65.33]    |
>
> **Experimental Setup**:
> - **Model**: 4-layer Multi-Layer Perceptron (MLP) with GELU activation.
> - **Input features**: 3072 input features for all hidden layers.
> - **Number of epochs**: 100.
> - **Dataset**: CIFAR-10.
> - **Optimizer**: Stochastic Gradient Descent (SGD) without momentum.
> - **Learning Rate**: Initial value \(1\mathrm{e-}1\), annealed using a learning rate scheduler.
> - **Batch Size**: 128.
> - **Data Augmentation**: Standard practices including random cropping, horizontal flipping, and normalization.
> - **Regularization**: Default weight decay set to 0.0 across all runs.
> - **Number of runs**: 10 runs per configuration.
>
> Our findings show that **Activation Decay (AD)**, which applies \(\ell_2\)-regularization to the penultimate activations, achieves comparable results to Baek et al. without the need for layer-wise tuning. Specifically, Baek et al.'s method requires tuning separate coefficients for intermediate layers (\(\sigma = 1\mathrm{e}{-3}\)) and for the last layer parameters (\(\sigma = 1\mathrm{e}{-2}\)). In contrast, our approach eliminates this complexity and achieves similar performance with a single hyperparameter, \(\sigma\), set to \(0.1\).
> The simplicity of our method reduces the number of hyperparameters to tune, making it both more practical and easier to analyze. Furthermore, regularizing the penultimate activation indirectly regularizes preceding layers, as the penultimate activation encapsulates their contributions. This insight highlights the theoretical and practical advantages of our approach.
> We add this experiement to appendix A.1 in the revised version of the paper.
>
> ## W2)
> "Activation decay can only be applied to the penultimate layer, whereas dropout can be applied in multiple locations within the backbone architecture. Therefore, the use of activation decay might be limited.
> "
> While activation decay is currently applied to the penultimate layer, this design choice is theoretically motivated by Theorem 2. The theorem demonstrates that the contributions from later layers dominate the curvature of the loss landscape, making the penultimate layer a critical target for regularization to achieve sharper reductions in sharpness and improve generalization.
> Moreover, our experimental results already compare activation decay to dropout on a variety of tasks. Specifically, in our MLP experiments on CIFAR-10 (Figure 2) and across all NLP tasks (Tables 2 and 5), activation decay consistently outperforms dropout.
>
> ## W3)
> "
> Figure 1a lacks annotations for better clarity.
> "
> Thank you we will add the missing legend for the figure.

---

> ### Author Response · Authors · 2024-11-18
>
> ## W4)
>
> "
> In Figure 1a, why is the estimated Hessian calculated only with respect to the final layer? Does this mismatch the theoretical results in Theorems 1 and 2?
> "
>
> We calculate the Hessian only with respect to the final layer to specifically measure the effect of activation decay, as derived in Theorem 3. This approach ensures we can directly evaluate the impact of the Hessian of the loss with respect to  z , the penultimate layer activations, which is the target of our regularization. By isolating the final layer, we effectively control the influence of the regularization, allowing for a more precise assessment of activation decay’s contribution.
> Furthermore, this setup provides a practical means of assessing the tightness of the bound presented in Theorem 1. Since Theorem 1 applies to all parameters, it is valid a fortiori for the parameters of the last layer alone. Thus, our approach both validates Theorem 3 and indirectly evaluates the applicability of Theorem 1 in the context of layer-specific regularization.
> We will update the text to make these points more explicit in the final version.
>
> ## W5)
>
> "
> In Table 1, this paper compares its methods with ASAM [Kwon et al., 2021] but is missing a citation for it. Why don’t the authors compare their methods with the original SAM?
> "
>
> We appreciate the reviewer highlighting the issue regarding the missing citation for ASAM. We used the implementation from the official PyTorch SAM repository ([GitHub link](https://github.com/davda54/sam/blob/main/example/train.py)), which, by default, sets `adaptive=True`. As a result, our comparison aligns with ASAM instead of the original SAM.
>
> We decided to retain ASAM for our experiments on this setting, as it is an updated version of SAM that incorporates adaptive scaling for improved performance. We will clarify this choice in the revised manuscript, include the correct citation for ASAM [Kwon et al., 2021], and explicitly mention that the results are based on ASAM rather than the original SAM.
>
> # Questions
> ## Q1)
> "
> In Table 2, could you explain why the optimal perturbation for your experiments with SAM is lower $\rho=0.15$ compared to the optimal  recommended for NLP tasks in Bahri et al. [2022]?
> "
>
> The best values for SAM in our experiments were selected from \(\rho = 0.01, 0.05, 0.1\). As indicated by the trend reported in Table 5, smaller values of \(\rho < 0.15\) consistently yield better results for our tasks. This observation motivated our choice of \(\rho\) values for the experiments. We will clarify this rationale in the paper to ensure transparency and provide additional context for the selection of \(\rho\).
>
>
> ## Q2)
>
> "
> The improvement of proposed methods in Section 4.2 is not significant. Could you extend your experiments to more complex datasets, such as CIFAR-100 and Tiny ImageNet, so that the differences and improvements are clearer?
> "
>
> We acknowledge the reviewer's observation regarding the significance of the improvements reported in Section 4.2. While our method demonstrates consistent improvements over baseline methods, we agree that evaluating on more complex datasets could better highlight its effectiveness.
>
> To this end, we have already included results on ImageNet with the MLP-Mixer architecture in Appendix A.2, which demonstrate the scalability and effectiveness of our approach on a larger and more complex dataset for vision tasks.

---

> > ### Comment · Reviewer_3SWC · 2024-11-26
> > **Thank you for your response**
> >
> > Thank you for your response. Most of my questions have been addressed.
> >
> > However, I still find the contribution of this paper to be limited. As noted, similar regularization ideas have already been explored in Baek et al. [2024]. Moreover, the connection between flatness and generalization remains controversial. The paper would benefit from being based on stronger assumptions beyond flatness and including more extensive experiments on vision models to better demonstrate the effectiveness of the proposed method.
> >
> > Therefore, I will maintain my current score.

---

> > > ### Author Response · Authors · 2024-11-26
> > >
> > > Thank you for your feedback.
> > >
> > > While we acknowledge similarities with Baek et al. [2024], our Activation Decay (AD) method is derived from Theorem 1 and Corollary 1, providing a principled justification for applying AD to the penultimate layer and weight decay (WD) to others. In contrast, Baek et al.’s approach relies on tuning AD coefficients across layers, introducing many hyperparameters in deep learning models with many layers. This added complexity can make their method less practical, particularly for deep architectures. Our method is straightforward and requires only one layer-specific tuning while achieving comparable or better results, thanks to its theoretical foundation.
> > >
> > > We agree that the connection between flatness and generalization is debated. Nonetheless, our validation is based on empirical results demonstrating improved generalization. We will work further to incorporate the directions suggested by the reviewer, exploring stronger assumptions beyond flatness and including more extensive vision model experiments in future work.

---

### Official Review · Reviewer_PhYr · 2024-11-02

**Soundness:** 1
**Presentation:** 3
**Contribution:** 2
**Rating:** 1
**Confidence:** 4

**Summary:**

Thank authors for submitting their work to ICLR 25'. The paper proposes so called $\textit{activation decay}$ (AD) regularizer ($\ell_2$ norm penatly of activations of the last layer) derived from Gaussian smoothing (convolution with Gaussian noise) designed to flatten sharp minima and improve generalization. It is argued that AD promotes the flatter minima and it is positioned as a less computational expensive alternative to stochastic regularizers such as dropout or SAM. The method is corroborated by experiments on CIFAR-10, InageNet and NLP tasks (BERT architecture).

**Strengths:**

S1: Timely, well suited topic for applications and ICLR community

**Weaknesses:**

While paper addresses the timely topic of generalization ability of deep learning it seems it does it by taking too many shortcuts as if it was written in a hurry. It proliferates into rather significant weaknesses in almost all sections, including errors in proof of the main results, as follows

W1 (Introduction, Related Work): The paper claims a theoretical contributions ($1^{st}$ bullet point in Introduction) as well. It stands on flattness = goog generalization premise (abstract, l:11 or l:28: "One of the key factors influencing generalization is the nature of the minima in the loss landscape.") or dedicated section 2.1. While this phenomenon has been reported under several settings experimentaly, flatness/curvature/Hessian is in general is parameter dependent and deep overparameterized networks are widely weight invariant, e.g., ReLU networks, pre-/post- non-lineartity rescaling, allowing for counter examples showing that sharp minima can generalize well and vice-versa, see for instance (Zhang et al., Understanding deep learning requires rethinking generalization, 2017).

More over that is dedicated section 2.1 that reviews the related work dedicated to flat minima, yet it does not mention any opposing works whatsoever, which seems to me a bit biased, given that this is to the best of my knowledge still an open problem (at theory at least). The paper should mention (Introduction, Related Work) these works and challenges and take necessary assumptions under which flatness implies good generalization to make theoretical part solid. In its current version the flatness is presented as the sufficient condition to generalization, which does not hold in general.

Section 3.
W2: The theory is presented for "near optimum" l171 "In the regime near a minimum, ...", Theorem 1, or even for "$\nabla \mathcal{L} = 0$" Theorem 2. Letting alone works as "Chaudhuri et al., Neural Network Weights Do Not Converge to Stationary Points: An Invariant Measure Perspective, PLMR 2022" questioning whether finding well generalizing solution requires convergence at all, presented Theorem 1 argues that flattening of landscape happens after converging to neighborhood of a particular optimum.

However, Theorem 1 gives an upper bound for smoothed (training) loss $\nabla_{\theta}^2 \mathcal{L}(\theta + \Delta)$, yet for generalization the $test$ loss, i.e., $without$ noise convolution (Gaussian smoothing), is relevant. Could authors present the argument leading to test loss improved bounds? Recall, under "near optimum" assumption the neighborhood of a local optimum $\theta_0$ is fixed ...

W3: Overall theoretical contribution:
W3a: Theorem 1. This is just a Theorem 2 taken from Dellattre as also mentioned (or its special case follows from a convolution with Gaussian noise). Proof in SM is also just 2 lines reference to Dellattre paper. I suggest not to present it as a standalone Theorem, but rather apply it on your settings.

W3b: Proof of Theorem 2 does not hold. Line 807: "At a local minimum, the gradient term involving the first derivatives vanishes." - unfortunately this does not imply that $\nabla_z \mathcal{L}=0$ because this is gradient w.r.t. outputs of the networks, not weights in general. In fact for convex loss $\nabla_z \mathcal{L}=0$ only vanishes in GLOBAL optimum. Thus the Eq. on line 809, does not hold and hence Theorem 2 conclusion is fault. (Btw. It also is mentioned not to yield any improvements in Discussion & COnclusion, line 497-499.)

Experiments
W4a. SAM. Exclusion of SAM from experiments, l 324, and it suboptimal results in Table 1 and 2. This is confusing to me as SAM is by definition method that explicitly finds flatter (and thus better generalizing optima according to paper) and weight decay or AD are just rougher (yet faster) "flat optima searching" methods. Thus while providing efficiency merits, they should not significantly overperform SAM. However this is not what is reported and to me it looks as a wrong hyperparameter choice for SAM. But then, are the experiments conclusive?
W4b. The difference in Table 1 between (AD + SAM) and SAM rows are not significant, given the standard deviation presented ...
W4c. Qute importantly, weight decay (WD) experiments with hyper parameter optimization are missing to be compared with AD. WD has all the computational and efficiency merits as AD and thus AD should outperform WD in accuracy to makes practical sense to replace it...

**Questions:**

See Weaknesses

**Details Of Ethics Concerns:**

As mentioned in Weaknesses, Theorem 1 is really (almost) a direct application of Theorem 2 from referenced Dellattre paper. It is mentioned by authors in the body of the paper, yet I m not sure it is ok to present Theorem 1 as a 'new' Theorem, in case the paper is accepted. I believe authors would mend this without issues in rebuttal, however.

---

> ### Author Response · Authors · 2024-11-18
>
> We thank the reviewer for their precious feedbacks, please find a revised version of the paper uploaded.
>
> # Weakness
> ## W1)
>
> "(Introduction, Related Work):
> The paper claims a theoretical contributions ( bullet point in Introduction) as well. It stands on flattness = goog generalization premise (abstract, l:11 or l:28: "One of the key factors influencing generalization is the nature of the minima in the loss landscape.") or dedicated section 2.1.
> While this phenomenon has been reported under several settings experimentaly, flatness/curvature/Hessian is in general is parameter dependent and deep overparameterized networks are widely weight invariant, e.g., ReLU networks, pre-/post- non-lineartity rescaling, allowing for counter examples showing that sharp minima can generalize well and vice-versa, see for instance (Zhang et al., Understanding deep learning requires rethinking generalization, 2017)."
> "More over that is dedicated section 2.1 that reviews the related work dedicated to flat minima, yet it does not mention any opposing works whatsoever, which seems to me a bit biased, given that this is to the best of my knowledge still an open problem (at theory at least). The paper should mention (Introduction, Related Work) these works and challenges and take necessary assumptions under which flatness implies good generalization to make theoretical part solid. In its current version the flatness is presented as the sufficient condition to generalization, which does not hold in general."
>
> Our primary goal is to develop a method that improves generalization by leveraging the concept of flat minima. While the relationship between flat minima and generalization is not theoretically guaranteed, there is empirical evidence suggesting a correlation between flatter minima and reduced generalization gaps (Jiang et al. 2020) . For instance, (Andriushchenko and Flammarion 2022)  and (Foret et al., 2021)  highlight the utility of flat minima in improving generalization through sharpness-aware methods.
>
> We validate the proposed Activation Decay (AD) method experimentally, demonstrating its effectiveness in enhancing generalization across various tasks. Our theoretical contributions focus on flat minima, where we use the empirical observation of their correlation with improved generalization to motivate and support the design of AD. The experiments confirm that the method achieves its goal of better generalization without degenerating into metrics tied solely to sharpness.
>
> To provide a balanced perspective, we have revised the manuscript to include references to opposing views in the Related Work section, acknowledging that flat minima do not universally imply better generalization and that the problem remains open in theory.
>
> Jiang, Y., Neyshabur, B., Mobahi, H., Krishnan, D., & Bengio, S. (2020). *Fantastic Generalization Measures and Where to Find Them*. Proceedings of the 8th International Conference on Learning Representations (ICLR 2020).
>
> Andriushchenko, M., & Flammarion, N. (2022). *Towards Understanding Sharpness-Aware Minimization*. Proceedings of the 39th International Conference on Machine Learning, PMLR 162, Baltimore, Maryland, USA.
>
> Foret, P., Kleiner, A., Mobahi, H., & Neyshabur, B. (2021). *Sharpness-Aware Minimization for Efficiently Improving Generalization*. Proceedings of the 9th International Conference on Learning Representations (ICLR 2021).

---

> ### Author Response · Authors · 2024-11-18
>
> ## W2)
>
> "The theory is presented for "near optimum" l171 "In the regime near a minimum, ...", Theorem 1, or even for $\nabla \mathcal{L} = 0$ Theorem 2. Letting alone works as "Chaudhuri et al., Neural Network Weights Do Not Converge to Stationary Points: An Invariant Measure Perspective, PLMR 2022" questioning whether finding well generalizing solution requires convergence at all, presented Theorem 1 argues that flattening of landscape happens after converging to neighborhood of a particular optimum."
>
> While (Chaudhuri et al., 2022) question whether finding well-generalizing solutions requires convergence, our theoretical framework assumes a setting where training achieves convergence to a local minimum with zero training error (Zhang et al., 2017) i.e interpolation regime. This is a common scenario in practice, particularly for over-parameterized models, where such convergence is facilitated by decreasing learning rate schedules.
>
> Theorem 1 specifically addresses the behavior of the smoothed loss landscape in the neighborhood of a minimum, aligning with the assumption that modern optimization methods (e.g., SGD with sufficient training epochs) often bring neural network weights close to such minima.
>
> It is important to note that our focus is on comparing the properties of the smoothed loss versus the regular loss to understand the regularization effect of Activation Decay (AD). This comparison establishes a framework for analyzing curvature and generalization effects in the vicinity of a minimum. Our smoothed loss can be cast as a base loss plus a regularization term on Hessian curvature, in the same vein as (Orvieto et al., 2022), where noise injection is used to improve generalization through implicit regularization. This formulation allows us to analyze the role of AD as a deterministic alternative to noise-based methods, providing similar generalization benefits while avoiding the instability associated with stochastic techniques.
>
> We do not, however, take into account the training dynamics of the smoothed loss itself, as analyzed in works such as (Nesterov et al., 2005). This distinction ensures that our analysis remains relevant to practical scenarios while acknowledging that non-convergent dynamics, as discussed by Chaudhuri et al., are beyond the scope of our current work. We have revised the discussion in Section 3 to clarify this scope, explicitly state our assumptions, and highlight the focus on the regularization effects of AD near a minimum.
>
> Chaudhuri, K., Das, P., & Lyu, K. (2022). *Neural Network Weights Do Not Converge to Stationary Points: An Invariant Measure Perspective*. Proceedings of the 39th International Conference on Machine Learning, PMLR 162.
>
> Zhang, C., Bengio, S., Hardt, M., Recht, B., & Vinyals, O. (2017). *Understanding deep learning requires rethinking generalization*. Proceedings of the International Conference on Learning Representations (ICLR).
>
> Orvieto, A., Kersting, H., Proske, F., Bach, F., & Lucchi, A. (2022). *Anticorrelated Noise Injection for Improved Generalization*. Proceedings of the 39th International Conference on Machine Learning, Baltimore, Maryland, USA, PMLR 162.
>
> Nesterov, Y. (2005). *Smooth minimization of non-smooth functions*. Mathematical Programming, 103(1), 127-152.
>
>
>
> "However, Theorem 1 gives an upper bound for smoothed (training) loss
> $\nabla^2_\theta \mathcal{L}(\theta + \Delta)$, yet for generalization the  loss, i.e.,  without noise convolution (Gaussian smoothing), is relevant. Could authors present the argument leading to test loss improved bounds? Recall, under "near optimum" assumption the neighborhood of a local optimum  $\theta_0$ is fixed ..."
>
> Theorem 1 provides an upper bound for the smoothed training loss. While it is true that generalization is ultimately evaluated on the unsmoothed test loss, we argue that the smoothed loss serves as a useful proxy for understanding the regularization effect of Activation Decay (AD).
>
> The smoothing operation can be interpreted as inducing an implicit regularization effect, where the smoothed loss is equivalent to the base loss plus a regularization term, similar to the analysis by (Orvieto et al., 2022). This regularization helps control the curvature of the loss landscape, promoting flatter minima that are empirically associated with improved generalization. By bounding the curvature of the smoothed loss, we indirectly regularize the curvature of the unsmoothed loss in the vicinity of the optimum \(\theta_0\).

---

> ### Author Response · Authors · 2024-11-18
>
> It is important to note that regularization terms are not considered during the evaluation of the test loss. The test loss is computed solely on the final learned parameters of the model without any additional penalties, as its purpose is to measure the model's performance on unseen data. This is a standard practice in machine learning (Goodfellow et al., 2016) to ensure fair evaluation and to accurately reflect the generalization ability of the trained model.
>
> Flatness is often studied in the context of the **optimization process itself**, where the goal is to understand the characteristics of minima in the training loss landscape. This provides insights into how the optimization trajectory interacts with the geometry of the loss surface, which is crucial for ensuring effective convergence to minima with favorable generalization properties.
>
> Orvieto, A., Kersting, H., Proske, F., Bach, F., & Lucchi, A. (2022). *Anticorrelated Noise Injection for Improved Generalization*. Proceedings of the 39th International Conference on Machine Learning, Baltimore, Maryland, USA, PMLR 162.
>
> Goodfellow, I., Bengio, Y., & Courville, A. (2016). *Deep Learning*. MIT Press.

---

> ### Author Response · Authors · 2024-11-18
>
> ## W3)
> Overall theoritical contribution
>
> ### W3)a)
>
>
> "Theorem 1. This is just a Theorem 2 taken from Dellattre as also mentioned (or its special case follows from a convolution with Gaussian noise). Proof in SM is also just 2 lines reference to Dellattre paper. I suggest not to present it as a standalone Theorem, but rather apply it on your settings."
>
> While Theorem 1 is adapted from prior work in the context of Randomized Smoothing from (Delattre et al, 2024), where noise is injected on the inputs of networks, our contribution lies in explicitly applying it to the gradient of the loss with respect to parameters, where the noise is injected directly into the parameters. While the technical proof itself is not novel, its application in our specific setting is. We will revise the paper to make this distinction clearer and state it as a corollary of Theorem 2 of Delattre et al. .
>
>
> ### W3)b)
>
> "Proof of Theorem 2 does not hold. Line 807: "At a local minimum, the gradient term involving the first derivatives vanishes." - unfortunately this does not imply that  $\nabla_z \mathcal{L} = 0$ because this is gradient w.r.t. outputs of the networks, not weights in general.
> In fact for convex loss $\nabla \mathcal{L}_z = 0$ only vanishes in GLOBAL optimum. Thus the Eq. on line 809, does not hold and hence Theorem 2 conclusion is fault.
> "
>
> Thank you for identifying this flaw in Theorem 2. We have updated the assumption in Theorem 2 to consider cases where the loss \(\mathcal{L}(\theta) = 0\), which is a reasonable assumption in over-parameterized deep neural networks that can fit the training data and achieve near-zero training loss (Zhang et al., 2017) . Under this assumption, \(\theta\) corresponds to a global minimum, ensuring that both \(\nabla_{\theta} \mathcal{L} = 0\) and \(\nabla_z \mathcal{L} = 0\). This revision makes the proof valid and aligns with the behavior of modern deep learning models during training. Under this corrected assumption the result and its implication remain the same.
>
>
> "
> (Btw. It also is mentioned not to yield any improvements in Discussion & COnclusion, line 497-499.)
> "
>
> The formula in Theorem 2 provides guidance for layer-specific regularization using either weight decay or activation decay. Since most configurations already used weight decay, we introduced activation decay as an additional regularization method, and empirical results confirmed its effectiveness. Activation decay directly regularizes the curvature of the loss with respect to \(z\), helping to control sharpness in the loss landscape. Our results demonstrated that combining weight decay with Hessian regularization of the loss with respect to \(z\) led to improvements over using either weight decay or Hessian regularization alone, in alignment with the guidance provided by Theorem 2.
>
> Zhang, C., Bengio, S., Hardt, M., Recht, B., & Vinyals, O. (2017). *Understanding deep learning requires rethinking generalization*. Proceedings of the International Conference on Learning Representations (ICLR).

---

> ### Author Response · Authors · 2024-11-18
>
> ## W4)
> Experiments
>
> ### W4)a)
> "SAM. Exclusion of SAM from experiments, l 324, and it suboptimal results in Table 1 and 2. This is confusing to me as SAM is by definition method that explicitly finds flatter (and thus better generalizing optima according to paper) and weight decay or AD are just rougher (yet faster) "flat optima searching" methods. Thus while providing efficiency merits, they should not significantly overperform SAM. However this is not what is reported and to me it looks as a wrong hyperparameter choice for SAM. But then, are the experiments conclusive?"
>
> The confusion arises because SAM theoretically aims to minimize worst-case sharpness, but in practice, as shown by (Wen et al. 2023), it primarily minimizes average sharpness due to its implementation. Weight decay, AD, or Adam (while less explicit), also reduce sharpness effectively, which can explain their competitive performance. Our method aligns with this observation, as it is also designed to minimize average sharpness, allowing it to compete effectively in generalization performance.
>
> Wen, K., Ma, T., & Li, Z. (2023). How Does Sharpness-Aware Minimization Minimize Sharpness? Published as a conference paper at ICLR 2023
>
> We adjust the scheduler for SAM, here are the obtained result for SAM for the experiment Classification with MLP on CIFAR-10:
>
> | SAM Rho Parameter            | Mean Test Accuracy (%) | 95% CI Lower Bound | 95% CI Upper Bound |
> |:-----------------------------|-------------------------:|---------------------:|---------------------:|
> | 0.0 (baseline)                         |                   63.03 |                62.81 |                63.25 |
> | 0.001                        |                   62.79 |                62.61 |                62.97 |
> | 0.002                        |                   63.01 |                62.72 |                63.30 |
> | 0.005                        |                   63.18 |                62.96 |                63.40 |
> | 0.007                        |                   63.14 |                62.86 |                63.42 |
> | 0.01                         |                   63.06 |                62.83 |                63.29 |
> | 0.02                         |                   63.24 |                63.02 |                63.46 |
> | 0.03                         |                   63.05 |                62.81 |                63.29 |
> | 0.04                         |                   63.10 |                62.89 |                63.31 |
> | 0.05                         |                   62.94 |                62.62 |                63.26 |
> | 0.06                         |                   62.74 |                62.45 |                63.03 |
> | 0.07                         |                   62.84 |                62.58 |                63.10 |
> | 0.08                         |                   63.00 |                62.75 |                63.25 |
> | 0.09                         |                   62.40 |                62.18 |                62.62 |
> | 0.1                          |                   62.39 |                62.10 |                62.68 |
> | 0.12                         |                   61.94 |                61.69 |                62.19 |
> | 0.15                         |                   60.70 |                60.48 |                60.92 |
> | 0.17                         |                   59.70 |                59.41 |                59.99 |
> | 0.2                          |                   59.47 |                59.17 |                59.77 |
> | **AD (ours) ($\sigma=0.1)$** |                   65.11 |                64.90 |                65.33 |
>
> **Note:** This table reports results for different values of \(\text{RhoSam}\) using the same configuration as described in the paper. The experiments were conducted using:
> - **Model**: 4-layer Multi-Layer Perceptron (MLP) with GELU activation.
> - **Input features**: 3072 features for all hidden features.
> - **Number of epochs**: 100
> - **Dataset**: CIFAR-10.
> - **Optimizer**: Stochastic Gradient Descent (SGD) without momentum.
> - **Learning Rate**: Initial learning rate set to \(1\mathrm{e-}1\) with annealing using a learning rate scheduler.
> - **Batch Size**: 128.
> - **Data Augmentation**: Standard data augmentation including random cropping and horizontal flipping and normalization.
> - **Regularization**: Weight decay set to 0.0 for all runs.
> - **Number of runs**: 10 runs for each configuration.
>
> We have updated Figure 2 in the revised version of the paper to include SAM results. Please find the code to reproduce table in zip file uploaded.

---

> ### Author Response · Authors · 2024-11-18
>
> ### W4)b)
> To clarify, while SAM performs exceptionally well on convolutional architectures (e.g., CNNs), its effectiveness appears to be reduced on dense-layer architectures.
> 1.	Computational Overhead: SAM requires additional computations, specifically the calculation of the sharpness of the model at each training step. This involves computing the loss and gradients for perturbed models, which can be expensive, especially for large models like LLMs. The overhead of perturbing the model and re-evaluating it can significantly slow down training, which is a concern for large-scale models that already require considerable computational resources.
> 	2.	Memory Requirements: SAM involves maintaining multiple versions of the model parameters (i.e., one for the perturbed version and one for the original), which can increase memory usage. Given the size of LLMs, this added memory requirement can become a limiting factor, making SAM less attractive for models with billions of parameters.
>
> ### W4)c)
> To clarify, our method does not aim to replace weight decay (WD) but rather combine Adaptive Decay (AD) with WD, leveraging guidance from Theorem 2 to complement their strengths. While WD is computationally efficient and effective as a regularizer, AD introduces dynamic adjustments that target improved flatness and generalization beyond what static WD can achieve. Indeed, AD includes input proagation based regularization whereas WD does not include this type of regularization.
>
> Importantly, the baselines in Table 1 already include weight decay parameters optimized through hyperparameter search to ensure a fair comparison. Our results demonstrate that combining AD with WD consistently enhances performance, showing that AD can outperform WD alone in accuracy, particularly when used in tandem with other regularization strategies. This combined approach makes a strong case for adopting AD in settings where computational resources allow for its additional benefits.

---

> ### Comment · Reviewer_PhYr · 2024-11-26
> **After rebuttal comments**
>
> Thanks to authors for their replies and addressing several raised issues to a certain extend. However, not fully in my view. And my score proposal stays unchanged.
>
> For instance, a correction of the proof of Th.2. (now assuming convergence to global optimum) brings rather a strong assumption raising further questions about applicability of the Theorem. Strictly speaking, noise injection will ensure, that $\mathcal{L} =0$ will be never attained exactly but stay in a vicinity of the global optimum at best ... So, theoretically speaking, Th 2. is not directly applicable to AD training and an 'approximative' version could be developed (using Lipschitzity adn so on, perhaps ...). I encourage authors to do a thorough revision and submit again.

---

### Official Review · Reviewer_X2Cr · 2024-11-03

**Soundness:** 2
**Presentation:** 3
**Contribution:** 2
**Rating:** 3
**Confidence:** 4

**Summary:**

The paper proposes a deterministic regularizer to ensure that training converges to a wider minima known to lead to improved generalization. Some experiments are conducted on computer vision and NLP tasks to validate the claim. The regularizer is also justified theoretically.

**Strengths:**

The paper clarity is good and the motivation to move away from stochastic regularization is important.

**Weaknesses:**

**Missing references**:
- Large Margin Deep Networks for Classification
- Minimizing Layerwise Activation Norm Improves Generalization in Federated Learning (closest)

**Missing experiments**:
- SAM with much finer set of p needs to be provided for **all** the experiments
- additional datasets and architectures need to be explored at least for the computer vision domain (as per the SAM paper)
- additional comparisons with dropout, stochastic depth needs to be provided

**Questions:**

In addition to the above concerns that need to be addressed. A lot of claims from the paper feel subjective, e.g., the batch size argument (in contrast to that paper finding (Stochastic Training is Not Necessary for Generalization). Also, the authors need to provide much more experimental details and configurations used for training as the work is currently not reproducible e.g. there is no mention of data augmentation in the entire paper.

Because of the above + all my concerns and the fact that this work provides a method that was already explored before and fails to expand on their empirical validation (and the derivations in the paper are well known and were already used in previous papers), I do not think there is novelty or insights to meet the acceptance level.

---

> ### Author Response · Authors · 2024-11-18
>
> We thank the reviewer for their precious feedbacks, please find a revised version of the paper uploaded.
>
> ## Weaknesses
>
> - **Missing References**:
>   - We appreciate the reviewer pointing out relevant works on related methods. We have included references to *Large Margin Deep Networks for Classification* and *Minimizing Layerwise Activation Norm Improves Generalization in Federated Learning*, which closely aligns with our method’s goals. These references further contextualize AD’s focus on generalization and link to margin-based improvements.
>
> - **Missing Experiments**:
>   - **SAM with finer \(\rho\)**: We acknowledge the importance of benchmarking SAM across a finer set of \(\rho\) values. We have conducted additional experiments with a range of \(\rho\) values and ensured fair comparisons.
>
> We adjust the scheduler for SAM, here are the obtained result for SAM for the experiment Classification with MLP on CIFAR-10:
>
> | SAM Rho Parameter            | Mean Test Accuracy (%) | 95% CI Lower Bound | 95% CI Upper Bound |
> |:-----------------------------|-------------------------:|---------------------:|---------------------:|
> | 0.0 (baseline)                         |                   63.03 |                62.81 |                63.25 |
> | 0.001                        |                   62.79 |                62.61 |                62.97 |
> | 0.002                        |                   63.01 |                62.72 |                63.30 |
> | 0.005                        |                   63.18 |                62.96 |                63.40 |
> | 0.007                        |                   63.14 |                62.86 |                63.42 |
> | 0.01                         |                   63.06 |                62.83 |                63.29 |
> | 0.02                         |                   63.24 |                63.02 |                63.46 |
> | 0.03                         |                   63.05 |                62.81 |                63.29 |
> | 0.04                         |                   63.10 |                62.89 |                63.31 |
> | 0.05                         |                   62.94 |                62.62 |                63.26 |
> | 0.06                         |                   62.74 |                62.45 |                63.03 |
> | 0.07                         |                   62.84 |                62.58 |                63.10 |
> | 0.08                         |                   63.00 |                62.75 |                63.25 |
> | 0.09                         |                   62.40 |                62.18 |                62.62 |
> | 0.1                          |                   62.39 |                62.10 |                62.68 |
> | 0.12                         |                   61.94 |                61.69 |                62.19 |
> | 0.15                         |                   60.70 |                60.48 |                60.92 |
> | 0.17                         |                   59.70 |                59.41 |                59.99 |
> | 0.2                          |                   59.47 |                59.17 |                59.77 |
> | **AD (ours) ($\sigma=0.1)$** |                   65.11 |                64.90 |                65.33 |
>
> **Note:** This table reports results for different values of \(\text{RhoSam}\) using the same configuration as described in the paper. The experiments were conducted using:
> - **Model**: 4-layer Multi-Layer Perceptron (MLP) with GELU activation.
> - **Input features**: 3072 features for all hidden features.
> - **Number of epochs**: 100
> - **Dataset**: CIFAR-10.
> - **Optimizer**: Stochastic Gradient Descent (SGD) without momentum.
> - **Learning Rate**: Initial learning rate set to \(1\mathrm{e-}1\) with annealing using a learning rate scheduler.
> - **Batch Size**: 128.
> - **Data Augmentation**: Standard data augmentation including random cropping and horizontal flipping and normalization.
> - **Regularization**: Weight decay set to 0.0 for all runs.
> - **Number of runs**: 10 runs for each configuration.
>
>
> We have updated Figure 2 in the revised version of the paper to include SAM results. Please find the code to reproduce table in zip file uploaded.
>
>
>   - For NLP tasks, the best values for SAM in our experiments were selected from \(\rho = 0.01, 0.05, 0.1\). As indicated by the trend reported in Table 5, smaller values of \(\rho < 0.15\) consistently yield better results for our tasks. This observation motivated our choice of \(\rho\) values for the experiments. We will clarify this rationale in the paper to ensure transparency and provide additional context for the selection of \(\rho\).

---

> ### Author Response · Authors · 2024-11-18
>
> - Furthermore, we clarify that the SAM experiments presented use Adaptive SAM (ASAM) (Kwon et al, 2021), with the best \(\rho\) value from [this reference](https://github.com/davda54/sam/issues/37) as validated by the authors.
>
>    Kwon, J., Park, H., Kim, J., & Choi, I. (2021). *ASAM: Adaptive sharpness-aware minimization for scale-invariant learning of deep neural networks*. Advances in Neural Information Processing Systems
>
>   - **Additional Datasets and Architectures**: While we focus primarily on common benchmarks for clarity (CIFAR10, ImageNet), we agree that additional datasets and architectures are valuable for validation, especially in the computer vision domain. Future work will explore AD across various architectures and datasets to extend empirical support.
>   We have  included results on ImageNet with the MLP-Mixer architecture in Appendix A.1, which demonstrate the scalability and effectiveness of our approach on a larger and more complex dataset for vision tasks.
>
>  -**Comparisons with Dropout and Stochastic Depth**: We acknowledge the importance of benchmarking against other regularization methods such as dropout and stochastic depth. In our current work, we provide comparisons with dropout on MLP classification tasks on CIFAR-10 (section 4.2) and on NLP tasks (section 4.3). While these comparisons establish the effectiveness of Activation Decay (AD) in diverse domains, including computer vision and NLP, we recognize that additional comparisons with stochastic depth across these tasks could further strengthen our findings. We plan to expand these comparisons in future iterations.
>
> ## Questions
>
> - **Subjectivity of Claims and Batch Size Argument**:
> Our batch size argument is based on findings from Keskar et al., whose sharpness-based analysis empirically shows that large-batch training leads to sharp minima, thus connecting batch size to the sharpness of the loss landscape. l.112 "A pivotal work on sharpness-based analysis by (Keskar et al.2016), empirically shows that large-batch training leads to sharp minima,
> making generalization worse than small-batch training, showing that flat minima tends to lead to better generalization."
> We will also include the different perspective provided by the reviewer, referencing *Stochastic Training is Not Necessary for Generalization*, to further clarify and balance the discussion on batch size and generalization.
>
> - **Experimental Details and Reproducibility**: In the revised version, we have added comprehensive training details, including data augmentation methods and batch sizes, to ensure reproducibility.
>
> ## Addressing Novelty Concerns
>
> While previous works explore similar regularization objectives, our approach uniquely focuses on deterministic, activation-based regularization (Activation Decay) in the penultimate layer, offering an efficient, low-variance alternative to traditional stochastic methods. This approach allows AD to consistently flatten sharp minima and enhance generalization, supported by empirical results and theoretical bounds on the loss curvature. The demonstration of Theorem 1 in the context of activation decay provides new insights, particularly as it was previously applied in the randomized smoothing (RS) context but is now used in a more general setting for regularizing sharp minima.
>
> We believe these adjustments address the reviewer’s concerns and highlight AD’s novel contribution as a deterministic alternative to stochastic regularization techniques. We appreciate the thorough review and suggestions for improvement.

---

> > ### Comment · Reviewer_X2Cr · 2024-11-28
> > **Answer to authors**
> >
> > I thank the authors for their additional experiments and comments. The provided novel results showcase no benefit of employing SAM which I find surprising and may be the product of other suboptimal hyper parameters as part of this experiment. This is to me a fundamental limitation as it is hard to judge as-is the applicability and generality of the method across numerous settings.
> > In light of the above and other reviewers' concerns I will keep my score for now.

---

### Official Review · Reviewer_1Xjs · 2024-11-04

**Soundness:** 2
**Presentation:** 3
**Contribution:** 2
**Rating:** 3
**Confidence:** 4

**Summary:**

This paper proposes an "activation decay" method to flatten sharp minima and thereby enhance generalization.

**Strengths:**

The authors conducted extensive experiments across CV and NLP tasks to demonstrate the effectiveness of activation decay (AD).

**Weaknesses:**

**Main Concern: Activation Decay (AD) Method**

- **Novelty**: AD appears to randomly permute parameters only in the last layer, which is akin to using average-direction SAM, i.e., $\min_{\theta}:\mathbb{E}_{g\sim N(0,I)}L(\theta+\rho g)$ [1][2], but applied solely to the last layer.

Additionally, related work [3] also suggests that SAM is not required for all parameters; applying it only to layer normalization layers suffices.

- **Theoretical supports:**
  - Theorem 1 applies to parameter permutation across all layers, unlike AD, which permutes parameters only in the last layer.
  - Theorem 2 could not illustrate why you only permunate the parameters in the last layer, as similar results should hold for other layers ($l\leq L-1$).
  - Theorem 3 provides only an upper bound for AD loss, insufficient to substantiate AD's effect similar to weight decay. A two-sided bound is needed for this claim.

**Secondary Concern: Practical Applicability**
- AD introduces an additional hyperparameter $\sigma$, requiring tuning for different tasks, which limits its flexibility. Results in Tables 1, 2, and 3 indicate that $\sigma$ values must be adjusted across tasks.


[1] Ujváry et al., Rethinking sharpness-aware minimization as variational inference. 2022.

[2] Wen et al., How sharpness-aware minimization minimizes sharpness? 2023.

[3] Müller et al., Normalization Layers Are All That Sharpness-Aware Minimization Needs. 2023.

**Questions:**

- Why do the authors refer to this method as "activation decay"? Note that it does not alter the activation function or activations directly, but only the parameters in the output layer.

- What would the experimental results be if parameters in layers other than the last layer were perturbed?

- In Table 1, why do the authors use an unconventional $\rho=2$ for SAM? The standard $\rho$ for SAM on CIFAR-10 is 0.05 or 0.1. Additionally, have the authors tuned SAM across all experiments to ensure fair comparison?

---

> ### Author Response · Authors · 2024-11-18
>
> We thank the reviewer for their precious feedbacks, please find a revised version of the paper uploaded.
>
> ## Novelty
> Activation Decay (AD) is not simply a parameter permutation method but a specific regularization on the penultimate layer activations, aiming to smooth the loss landscape by controlling sharpness in a manner similar to weight decay on activations rather than parameters. Unlike SAM or techniques that apply modifications to normalization layers alone, AD focuses on directly reducing the sharpness at critical points by regularizing activations. This distinction is crucial, as AD provides a deterministic and computationally efficient way to flatten minima, specifically targeting overfitting without requiring gradient-based perturbations across layers. This unique approach makes AD computationally more feasible, especially for large-scale tasks.
>
> ## Theoretical Support Concerns
>
> - **Theorem 1**: Injecting noise across all layers introduces significant variance, as reported in Orvieto et al. (2023) [1], which can be detrimental to stable training and generalization. Our method is deterministic, thus avoiding this added variance. By focusing regularization on the penultimate layer’s activations, our approach provides controlled regularization that avoids excess variance. Combined with weight decay, activation decay offers effective control over the overall curvature of the Hessian, as supported by Theorem~2.
>
> [1] Orvieto, A., Kersting, H., Proske, F., Bach, F., & Lucchi, A. (2023). *Anticorrelated Noise Injection for Improved Generalization*. Proceedings of the 40th International Conference on Machine Learning (ICML 2023).
>
>
> - **Theorem 2**: While it is true that Theorem 2 can be applied to all layers, our empirical results indicate that focusing on the penultimate layer yields effective curvature control and is computationally efficient. This approach is convenient as it allows for weight decay on earlier layers and activation decay on the penultimate layer, casting the regularization of the Hessian of the loss with respect to \( z \) as a deterministic form—activation decay.
> Applying activation decay specifically to the penultimate layer avoids the need for noise injection across all layers, which would otherwise lack a closed-form solution and be computationally expensive. By focusing on the penultimate layer, we achieve stable and efficient curvature control, as confirmed by our experimental results.
>
>
> - **Theorem 3 and Two-sided Bound**: The theorem provides an upper bound that approximates the effect of activation decay (AD) similarly to weight decay, specifically for the activations in the last layer. While a two-sided bound might capture the dynamics more fully, it is generally sufficient to minimize an upper bound to achieve the desired regularization effect. Empirically, we observed that this upper bound is effective in demonstrating the regularization effect by flattening minima and improving generalization. Additionally, the regularization translates clearly as a base loss plus a regularization term, making the formulation intuitive and straightforward to apply.
>
>
> ## Practical Applicability and Hyperparameter Tuning
>
> We acknowledge that AD introduces an additional hyperparameter (\(\sigma\)) for each task. Similar to SAM and weight decay, AD requires parameter tuning to achieve optimal performance. However, as shown in Tables 1, 2, and 3, \(\sigma\) adjustments are minimal and do not vary drastically across tasks, suggesting that tuning is not overly complex.
> Future work may focus on developing adaptive strategies for setting \(\sigma\) : one potential approach could involve setting \(\sigma\) proportional to the norm of the weights, allowing \(\sigma\) to scale adaptively with the model’s parameter magnitudes in the same flavour as AdaptativeSAM (Kwon et al. 2021).

---

> ### Author Response · Authors · 2024-11-18
>
> ## Questions
>
> - **Clarification on “Activation Decay” Terminology**: We refer to the method as “activation decay” because it directly regularizes the activations of the penultimate layer by applying an \(\ell_2\) norm regularization to the activations of the network’s input as it propagates through the layers. Unlike methods that modify parameters, AD regularizes the output activations, which influences the curvature of the loss landscape, justifying the term “activation decay”, please see l.240 in the paper. Since this regularization is applied to the activations, it is input-dependent and involves not only the parameters in the last layer but also all the parameters in preceding layers.
>
> - **Experiments with Layers Other Than the Last Layer**: In the current study, we chose to focus on the penultimate layer’s activations due to their critical influence on loss curvature, as well as for convenience, as stated previously. This choice allows for regularization that impacts both the final layer and all preceding layers, providing an effective and computationally feasible approach.
>
> - **Unconventional \(\rho\) for SAM in Table 1**: We acknowledge that there was a mistake in Table 1; we used Adaptive SAM (ASAM) instead of the conventional SAM. For ASAM, we have used the best \(\rho\) value as reported and validated by the SAM authors in [this reference](https://github.com/davda54/sam/issues/37). We apologize for the oversight and have ensured that all settings reflect the correct values for fair comparison. Please refer to Kwon et al. (2021) for more details on ASAM.
>
> Kwon, J., Park, H., Kim, J., & Choi, I. (2021). *ASAM: Adaptive sharpness-aware minimization for scale-invariant learning of deep neural networks*. Advances in Neural Information Processing Systems

---

> > ### Comment · Reviewer_1Xjs · 2024-12-02
> >
> > Thank you for the responses. However, my primary concerns remain unresolved, and I will maintain my initial score.
> >
> > For instance, I am still concerned about the novelty of the work. The smoothed final loss (Eq. (3)) appears identical to the formulation of applying average-direction SAM solely to the output layer. Furthermore, the clarifications provided for Q1, Q2, and Q3 are not satisfactory.

---

### Note · Authors · 2025-01-22

I have read and agree with the venue's withdrawal policy on behalf of myself and my co-authors.